# Cog-VADU: A Training-Free Cognitive Reasoning Framework for Video Anomaly Detection and Understanding

## Abstract

Video Anomaly Detection (VAD) aims to temporally localize abnormal events in videos. Most existing approaches rely on dataset-specific training and curated annotations, limiting generalization in open-set scenarios. Recent zero-shot methods based on Large Vision-Language Models (LVLMs) alleviate this dependency but often lack temporal continuity and structured reasoning. We propose **Cog-VADU**, a fully training-free framework that reformulates VAD as a sequential cognitive reasoning task. Cog-VADU introduces *Chain-of-Anomaly Detection Thought Prompting* (CoADTP), which unrolls an LVLM into a recurrent reasoning chain across video segments. By propagating structured rationales over time, the model maintains implicit temporal memory, enabling robust discrimination between complex anomalies and high-motion normal activities. To improve reliability, we further design a cross-modal re-ranking stage that aligns textual rationales with visual embeddings, enforcing semantic consistency and temporal coherence for refined and stable predictions. Extensive experiments on multiple public VAD benchmarks demonstrate that Cog-VADU achieves competitive zero-shot performance. Moreover, cross-model evaluations show that CoADTP consistently enhances reasoning-based anomaly detection in a model-agnostic manner, providing interpretable and generalizable anomaly understanding for real-world applications.

## 1 Introduction

Video surveillance plays a vital role in monitoring activities in public and private spaces. *Video Anomaly Detection (VAD)* Sultani et al. (2019) seeks to temporally localize unusual events such as violence, accidents, or security violations. Anomalies are typically characterized as deviations in appearance or motion from expected patterns Saligrama et al. (2010), often confined to short segments. However, abnormality is not purely statistical. Events may appear anomalous due to contextual mismatch rather than visual irregularity alone. A running crowd, for instance, may indicate panic in one setting and celebration in another. Such distinctions require situational understanding that extends beyond low-level motion cues.

***Early vision-only methods*** Liu et al. (2018); Liu & Ma (2019); Wu et al. (2020); Tian et al. (2021); Wan et al. (2021); Huang et al. (2022); Zhou et al. (2023a); Lv et al. (2023); Zhang et al. (2024a) primarily focus on modeling appearance and motion patterns to compute anomaly scores from video data. While these approaches have shown effectiveness in constrained or well-defined environments, they fundamentally lack semantic and contextual reasoning capabilities. Consequently, they are unable to explain *who* is involved, *what* is occurring, or *why* an event should be considered anomalous. The absence of high-level contextual understanding, an essential component for reliable video anomaly detection, significantly limits their robustness and generalization when deployed in complex real-world surveillance scenarios.

***Multimodal approaches*** Yuan et al. (2023); Wu et al. (2023); Chen et al. (2023); Yang et al. (2024); Tang et al. (2024); Ma et al. (2025); Yang et al. (2025) seek to overcome these limitations by incorporating multimodal cues through vision-language models (VLMs). By integrating visual observations with semantic and linguistic information, these methods improve detection performance and enhance interpretability, particularly in surveillance scenarios where visual cues alone are ambiguous or insufficient. Despite their promising results, the majority of these approaches rely on task-specific fine-tuning or instruction tuning

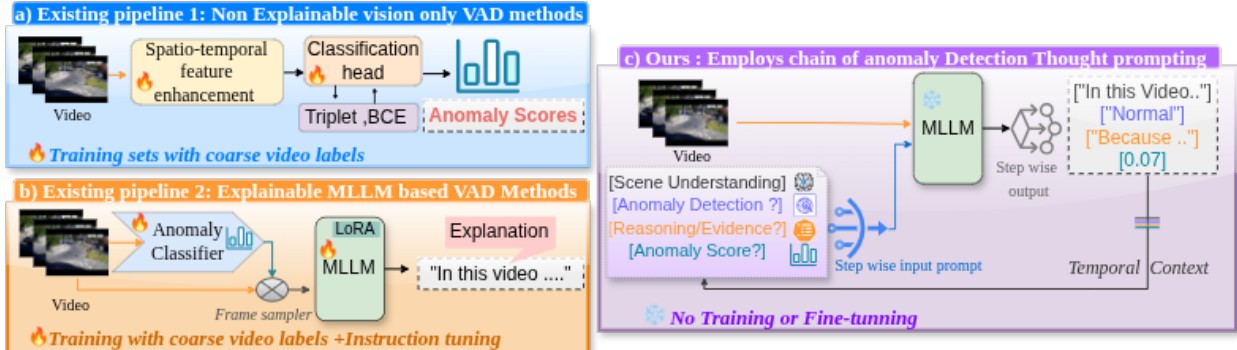

Figure 1: Comparison of video anomaly detection paradigms. (a) Vision-only methods rely on supervised training and produce anomaly scores without semantic reasoning. (b) Multimodal LLM-based VAD generates anomaly confidence and provides post-hoc explanations. (c) Cog-VADU employs a frozen LVLM with structured prompting and temporal cues to perform joint, training-free anomaly detection and reasoning.

using large-scale, carefully curated datasets, which are expensive to collect and difficult to scale. Such dependence on extensive training not only increases computational cost but also necessitates retraining when deployed in new environments or scenarios with shifting contextual distributions, thereby limiting scalability and adaptability, constraining open-set generalization.

To reduce the reliance on training, several training-free approaches, such as Zanella et al. (2024), exploit pretrained VLMs for caption generation and semantic matching, often employing models such as BLIP-2 Li et al. (2023a). Although these methods eliminate the need for additional training, they typically rely on shallow captioning or similarity-based matching strategies. As a result, they lack explicit multi-step reasoning mechanisms and struggle to capture complex contextual relationships, long-range temporal dependencies, and evolving scene dynamics. This often causes them to miss nuanced temporal and situational cues that are critical for distinguishing subtle anomalies from normal activities, particularly in cases requiring semantic interpretation and temporal causality.

In parallel, other approaches based on large vision-language models (LVLMs) Tang et al. (2024); Huang et al. (2025); Zhu et al. (2025); Ye et al. (2025) attempt to further exploit the inherent reasoning capabilities of such models to provide more interpretable anomaly detection, frequently by introducing explicit reasoning chains or language-based explanations. However, despite incorporating reasoning components, these methods still depend on task-specific training or supervision and, in some cases, specialized reasoning datasets with the requirement of high computational resources. This reliance restricts their ability to generalize across diverse datasets, unseen anomaly types, and real-world surveillance scenarios where annotations are limited or entirely unavailable.

**Key Findings.** Fundamentally, video anomaly detection requires *contextual reasoning* rather than relying solely on pattern deviation or statistical irregularities. Human judgment of anomalies involves understanding the overall scene, interpreting situational context, modeling temporal evolution, and reasoning about deviations from expected behavior before arriving at a final decision. Such reasoning enables the distinction between visually similar yet semantically different events, such as staged wrestling versus genuine acts of violence, or crowded yet orderly gatherings versus chaotic confrontations. Despite its critical importance, this reasoning-centric perspective remains largely underexplored in existing VAD methods, which predominantly depend on low-level visual deviations, heuristic thresholds, or naive prompting strategies that lack structured reasoning and temporal awareness. In light of these observations, we investigate the following unexplored yet fundamental question:

> *Can Video Anomaly Detection (VAD) be reformulated as a simple reasoning-driven, zero-shot detection and understanding problem by leveraging pretrained vision-language models, without any task-specific training or fine-tuning?*

We address this question with **Cog-VADU**, illustrated in Fig. 1. Cog-VADU treats anomaly detection as a structured cognitive reasoning process unfolding across video segments. A frozen, model LVLM is prompted to generate step-wise rationales that persist over time, forming an implicit temporal memory. Initial predictions are then refined through cross-modal rationale alignment, enforcing semantic consistency and suppressing hallucinations. The resulting system produces temporally coherent anomaly scores along with interpretable, glass-box explanations without any fine-tuning.

**Contributions.**

- We present **Cog-VADU**, a training-free formulation of VAD that models anomaly detection as sequential cognitive reasoning and understanding over video segments using pretrained LVLMs.

- We propose *Chain-of-Anomaly Detection Thought Prompting (CoADTP)*, enabling step-wise rationale propagation and temporally coherent zero-shot inference.

- We introduce a cross-modal re-ranking rationale refinement stage that aligns textual reasoning with visual representations to stabilize predictions.

- We provide unified re-annotations of UCF-Crime and XD-Violence that account for causal and temporal relationships, reducing label inconsistencies and enabling fair evaluation.

- Experiments across benchmarks demonstrate competitive zero-shot performance alongside interpretable reasoning outputs.

## 2 Related Work

**Video Anomaly Detection (VAD).** Video Anomaly Detection (VAD) aims to identify frames or temporal segments that deviate from normal behavior Hasan et al. (2016), and is fundamental to public safety Sultani et al. (2019) and video understanding Wu et al. (2020). Traditional VAD methods predominantly rely on deep neural networks using visual cues, achieving strong performance but lacking semantic understanding and interpretability. Existing methods are commonly categorized as Non-Explaniable, such as unsupervised, weakly supervised, and fully supervised and Explainable like MLLM-VADs.

***Unsupervised Methods.*** Unsupervised VAD Liu et al. (2018); Gong et al. (2019); Wang & Cherian (2019); Thakare et al. (2022); Tur et al. (2023); Yang et al. (2023) learn normal patterns from unlabeled or normal-only data and detects anomalies as deviations. While effective for modeling regular behavior, they struggle with subtle anomalies and generalization to unseen environments due to the absence of anomaly priors.

***Fully Supervised Methods.*** Fully supervised approaches Wang & Miao (2010); Liu & Ma (2019) utilize frame-level annotations for both normal and abnormal events, enabling precise localization but requiring extensive manual labeling, which limits scalability.

***Weakly Supervised Methods.*** Weakly supervised VAD Sultani et al. (2019); Tian et al. (2021); Lv et al. (2023) leverages video-level labels, striking a balance between annotation cost and performance. The dominant paradigm in this setting is Multiple Instance Learning (MIL), first introduced to VAD by Sultani et al. (2019), where a video is treated as a bag of segments, and a positive (anomalous) bag is assumed to contain at least one anomalous segment, while a negative bag contains none. Building on this formulation, subsequent works have progressively improved different components of the MIL framework. These include enhanced instance weighting Ilse et al. (2018), improved feature magnitude modeling Chen et al. (2022), as well as advances in efficiency and temporal modeling Gao & Liu (2025). More recent efforts further address robustness to label noise and uncertainty Zhou et al. (2023b;a).

**Multimodal and Explainable VAD.** To address the lack of semantic understanding, recent methods Yuan et al. (2023) integrate vision-language models (VLMs) to enable explainable video anomaly detection. Early approaches such as Zhang et al. (2024b) and its extension Zhang et al. (2025b) combine temporal sampling Zhou et al. (2023a) with VLM-based reasoning to generate interpretable anomaly descriptions.

Zero-shot and multimodal methods have further explored pretrained VLMs. Zanella et al. (2024) employs BLIP-2 Li et al. (2023a) and LLaMA-2 for caption-based anomaly scoring using task-specific prompts. VAD-LLaMA Lv & Sun (2024) introduces instruction tuning and temporal memory for weakly supervised detection. CLIP-based approaches such as VadCLIP Wu et al. (2023) perform frozen vision–language alignment enhanced with adaptive prompts and MIL-style objectives. VERA Ye et al. (2025) performs coarse-to-fine anomaly reasoning via scene-temporal integration, but relies on verbal reasoning training and exhibits limited generalization.

Other multimodal extensions include TEVAD Chen et al. (2023), which fuses visual and textual representations with multi-scale temporal modeling for caption-enhanced anomaly reasoning. Similarly, Pu et al. Pu et al. (2024) propose Prompt-Enhanced Learning (PEL) and Temporal Context Aggregation (TCA) to refine semantic alignment, reuse similarity matrices efficiently, and improve discriminability via knowledge-guided prompts, but still depend on training or dataset-specific design choices.

**Chain-of-Thought and Reasoning.** Chain-of-thought (CoT) prompting Zhang et al. (2022); Wei et al. (2023) has demonstrated that providing exemplars with intermediate reasoning steps elicits complex reasoning in sufficiently large language models, enhancing performance on arithmetic, common sense, and symbolic tasks. Inspired by CoT prompting in text-based LLMs. Recent studies extend CoT to multimodal and video understanding to focus on generating intermediate reasoning steps for visual understanding Zhang et al. (2024c; 2025c); Fei et al. (2024); Arnab et al. (2025), yet its application to video anomaly detection remains largely unexplored. Most existing VAD methods do not integrate structured reasoning with temporal feedback, leaving a gap between reasoning-driven prompting and practical surveillance analysis.

**Summary.** These limitations highlight the need for a training-free, zero-shot video anomaly detection framework that can fully exploit the intrinsic world knowledge, understanding, and multimodal reasoning capabilities of large vision-language models, without relying on fine-tuning or instruction-tuned datasets.

# 3 Our Methodology: Cog-VADU

We introduce **Cog-VADU**, a two-stage training-free, cognitive reasoning-driven framework for video anomaly understanding and detection, built upon a frozen large vision-language model (LVLM). Cog-VADU performs anomaly localization, explanation, and structured cognitive reasoning in a fully zero-shot manner. Our approach requires no fine-tuning, adapter modules, spatio-temporal backbones, or task-specific training. In the following, we first formalize the video anomaly detection problem and then describe the design of our reasoning-driven framework.

## 3.1 Problem Formulation

Consider a test video $V = \{I_1, I_2, \ldots, I_T\}$, where $I_t$ is the $t$-th frame and $T$ is the total number of frames. Our goal is to predict a frame-level anomaly score sequence

$$\hat{A} = [\hat{a}_1, \ldots, \hat{a}_T], \quad \hat{a}_t \in [0, 1], \tag{1}$$

where higher values indicate greater abnormality.

Unlike conventional methods, Cog-VADU does not require labeled training data. Using a frozen, model-agnostic LVLM, it performs zero-shot inference while maintaining temporal coherence across frames. For each segment, the framework outputs not only anomaly scores but also interpretable semantic descriptions and structured reasoning rationales, forming a glass-box explanation grounded in visual content and temporal context.

## 3.2 How Effective Is Reasoning-Based Prompting for VAD?

Prompting is central to exploiting large vision-language models (LVLMs), as it defines task intent and guides inference. Its effectiveness depends on how well task-specific knowledge and reasoning structures are encoded.

Prior work highlights the impact of prompt design for anomaly detection. For instance, Ye et al. (2025) learn task-specific prompts while keeping the backbone model frozen, but requires dataset-specific optimization

and additional training. In contrast, Zanella et al. (2024) uses lightweight caption-based prompting, which is efficient but lacks explicit task decomposition and structured reasoning.

We evaluate different prompting strategies to assess their effectiveness. A naive captioning prompt (Prompt 1) produces fluent descriptions but weak anomaly discrimination due to the lack of task guidance. Adding structured prompts with anomaly categories and confidence scores (Prompt 2) improves performance by enforcing task awareness, as shown in Fig. 2. However, predictions remain unstable, reflected in lower Acc@0.5 and AP, as the model overemphasizes isolated cues without contextual severity. Moreover, such prompts lack (i) step-wise reasoning, (ii) continuous severity modeling, and (iii) temporal consistency across segments.

To address these limitations, we propose *Chain-of-Anomaly Detection Thought Prompting* (CoADTP), which introduces structured reasoning, severity-aware scoring, and temporal feedback via previous segments. As shown in Table 1, CoADTP achieves the best performance, demonstrating that explicit reasoning with temporal continuity is critical for robust VAD.

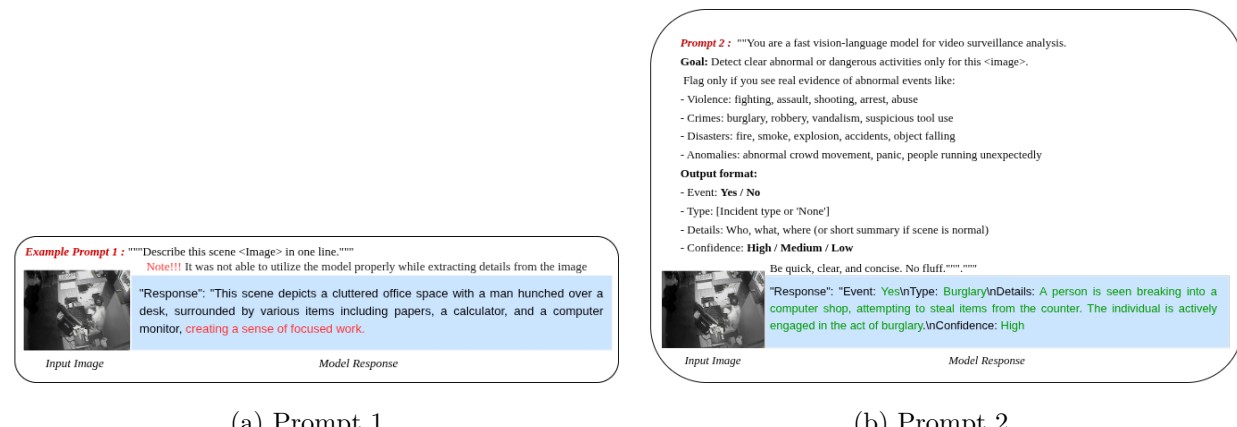

(a) Prompt 1      (b) Prompt 2

Figure 2: Comparison of prior-aware prompts for zero-shot VAD.

### 3.3 CoADTP: Chain of Anomaly-Detection Thought Prompt

To fully exploit the reasoning capabilities of LVLMs, we reformulate video anomaly detection as a training-free, prompt-driven reasoning task. Inspired by chain-of-thought prompting Zhang et al. (2022); Wei et al. (2023); Fei et al. (2024); Arnab et al. (2025), we introduce **Chain-of-Anomaly Detection Thought Prompting (CoADTP)**, which integrates anomaly priors and temporal context within a unified prompting framework.

The prompt is designed to enforce a structured reasoning pipeline prior to prediction. Given a video clip $v_i$, the LVLM first performs scene understanding by identifying objects, actions, and interactions. It then incorporates semantic priors from the anomaly database together with contextual information from the previous clip. Conditioned on this combined information, the model generates structured reasoning outputs, including observations, a concise description, anomaly assessment, and a rationale explaining the decision. Finally, it evaluates the severity of the event and produces a calibrated anomaly score.

Formally, CoADTP guides a frozen LVLM to generate outputs through three complementary components: (i) step-wise chain-of-thought reasoning, (ii) a lightweight text-based anomaly database encoding prior knowledge of normal and abnormal events, and (iii) temporal feedback across consecutive clips for score stability. These components enable consistent anomaly scoring and interpretable predictions without any additional training.

In contrast, CoADTP directly exploits the intrinsic reasoning capability of foundation-scale VLMs through structured prompting. The model is guided to sequentially *describe*, *analyze*, *decide*, and *justify/score* each event using prior knowledge and temporal context. This transforms VAD into an interpretable, context-

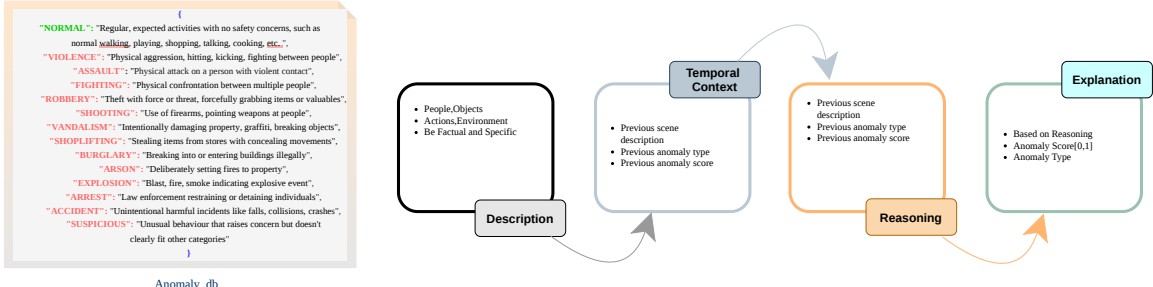

Figure 3: Workflow of CoADTP integrating anomaly priors and temporal reasoning cues.

aware inference process capable of capturing causal relations, temporal dependencies, and situational nuances entirely without training.

**Design components.**

- **Step-wise (chain-of-thought) reasoning.** Instead of directly predicting abnormality, the model follows a fixed reasoning chain:

$$observe \rightarrow describe \rightarrow reason \ \& \ assess \ severity \rightarrow decide \ category \rightarrow score.$$

  The final anomaly score lies in $[0, 1]$, where 0 indicates normal and 1 indicates abnormal behavior. This decomposition breaks down complex judgments into interpretable sub-steps (Fig. 3, panel (b)).

- **Anomaly text database (zero-shot prior).** To inject task-specific knowledge without training, we introduce a compact *verbal anomaly database* containing anomaly class names and concise definitions of normal and abnormal patterns. By explicitly referencing this database in the prompt, the LVLM can semantically align visual evidence with anomaly categories and output both an anomaly *type* and a concise rationale (Fig. 3, panel (a)). This eliminates the need for dataset-specific fine-tuning or annotation-heavy preparation.

- **Temporal feedback for stability.** To ensure temporal consistency, the prompt $P_i$ for clip $v_i$ is conditioned on the structured contextual rationale of the previous clip:

$$R_{i-1} = \big(\text{desc}_{i-1}, \ \text{reason}_{i-1}, \ \text{type}_{i-1}, \ a_{i-1}\big), \tag{2}$$

$$P_i = \mathcal{P}(v_i, R_{i-1}), \tag{3}$$

  where $a_{i-1}$ is the anomaly score. This recursive conditioning reduces prediction noise and improves temporal stability.

This design leads to improved score stability and detection accuracy compared to simple prompting strategies (see Sec. 4).

### 3.4   Raw Stage

Our methodology follows a two–stage design, as shown in Fig. 4. The *raw stage* produces zero-shot, segment-level anomaly scores and concise textual rationales using a vision–language model (LVLM), guided by our CoADTP prompt. No additional heads, fine-tuning, or parameter updates are used. *Segmentation and sampling.* Let a video consist of $F$ frames $V = \{I_i\}_{i=1}^{F}$, where $I_i$ denotes the $i$-th frame. We define segment centers at a fixed stride $d$, yielding

$$\mathcal{C} = \{I_{(u-1)d+1} \mid u = 1, \ldots, h\}, \quad h = \left\lfloor \frac{F}{d} \right\rfloor. \tag{4}$$

Table 1: Ablation of prompting strategies in the RAW stage on UCF-Crime. Prompt 1 is a minimal caption-style prompt; Prompt 2 is a format-structured surveillance prompt with anomaly categories and an output template but no reasoning; CoADTP adds step-wise reasoning, an anomaly database, and temporal feedback.

| Prompt | Prior | Reason | Temporal | AUC ↑ | AP ↑ | Acc@0.5 ↑ |
|---|---|---|---|---|---|---|
| Prompt 1 (Minimal) | ✗ | ✗ | ✗ | 0.7111 | 0.2150 | 78.28 |
| Prompt 2 (Format-Structured) | ✓ | ✗ | ✗ | 0.7403 | 0.2119 | 57.93 |
| CoADTP(Raw Stage) | ✓ | ✓ | ✓ | **0.7888** | **0.3773** | **82.41** |

For each center frame $I_{(u-1)d+1}$, we construct a temporal window of length $T$ seconds (e.g., $T = 10\text{s}$) centered around it. Assuming a frame rate of $r$ frames per second, this corresponds to a window of $Tr$ frames. From this window, we uniformly sample $m = 10$ frames to form the segment $V_u$. Prompted VLM inference. For each segment $u$, we form a CoADTP prompt $P_u$ that enforces a step-wise analysis (describe → reason → decide → explain) and references a lightweight anomaly *text* database; it includes temporal feedback from the previous segment (summary and score). With frozen $f_{\text{LVLM}}$ parameters $\theta$, we query

$$(\text{desc}_u,\ \text{reason}_u,\ \text{type}_u,\ a_u) = f_{\text{LVLM}}(V_u;\ \theta,\ P_u), \tag{5}$$

where $\text{desc}_u$ is a two-sentence clip description, $\text{reason}_u$ is a brief rationale, $\text{type}_u$ is a single anomaly category (e.g., `NORMAL`, `FIGHTING`, `ACCIDENT`), and $a_u \in [0, 1]$ is the raw anomaly score (higher is more anomalous).

**Outputs of the raw stage.** Repeating equation 5 for all segments yields

$$\begin{aligned}
\tilde{\mathbf{A}} &= [a_1, \ldots, a_h], & \mathbf{T} &= [\text{type}_1, \ldots, \text{type}_h], \\
\mathbf{D} &= [\text{desc}_1, \ldots, \text{desc}_h], & \mathbf{R} &= [\text{reason}_1, \ldots, \text{reason}_h].
\end{aligned} \tag{6}$$

These raw scores and textual outputs are passed to the refinement stage, where we stabilise and sharpen localisation and inject global scene context using cross-model context retrieval (Sec. 3.5). Quantitative analyses of the raw-stage behaviour for different prompting strategies are reported in the Table 1.

**Prompt Sensitivity Analysis.** To verify that Cog-VADU's raw-stage performance is driven by the *structural* properties of CoADTP rather than by specific lexical choices, we evaluate three prompts spanning two qualitatively different levels of structure (Table 1):

- **Prompt 1 (Minimal):** A one-line caption instruction ("Describe this scene and generate an anomaly score between 0 and 1") with no anomaly priors, no reasoning, and no temporal context.

- **Prompt 2 (Format-Structured):** A surveillance-tailored prompt providing *surface* structure bulleted anomaly categories (Violence, Crimes, Disasters, Anomalies) and a defined output format (Event / Type / Details / Confidence) but lacking step-wise reasoning or temporal context.

- **CoADTP (Reasoning-Structured):** The full chain-of-anomaly-detection prompt with explicit step-wise reasoning, a richer anomaly database, and temporal feedback across segments.

The gap between Prompt 1 and Prompt 2 is small ($+2.92\%$ AUC, $-0.31\%$ AP), indicating that adding *format-level* structure alone categories and output templates yields only marginal gains. In contrast, CoADTP outperforms both ($+7.77\%$ AUC and $+16.23\%$ AP over Prompt 1) by introducing *reasoning-level* structure: step-wise analysis and temporal propagation. Performance scales monotonically with the *depth* of structural complexity rather than surface wording. A component-wise breakdown of which sub-mechanisms (reasoning chain, anomaly database, temporal feedback, cross-modal re-ranking) drive these gains is provided in Table 9 (Appendix C). This robustness is further confirmed by our cross-backbone results (Sec. 4.1), where CoADTP improves multiple LVLM backbones (Video-ChatGPT, Video-LLaVA, LLaVA-Next-Video, QwenVL2, InternVL2, Video-R1) by large margins *without* prompt re-tuning, indicating that the prompt structure transfers robustly across model families.

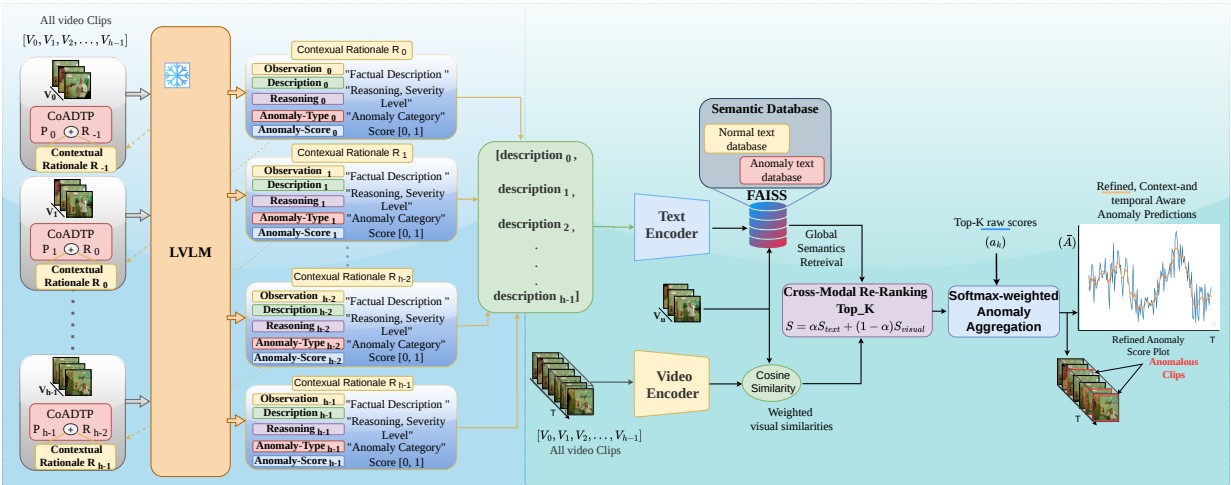

Figure 4: Overview of **Cog-VADU**, a training-free, reasoning-driven video anomaly detection framework. Videos are divided into clips and processed by a frozen $f_{\text{LVLM}}$ using Chain-of-Anomaly-Detection Prompting (CoADTP) to produce anomaly scores and textual rationales. A refinement stage employs **ImageBind** embeddings to retrieve and re-rank semantically related visual-text contexts, yielding final contextualized anomaly scores via softmax-weighted similarity aggregation.

## 3.5 Refinement Stage: Cross-Modal Re-Ranking

The raw stage produces independent segment-level anomaly scores $\tilde{\mathbf{A}} = [a_1, \ldots, a_h]$, which may lack global semantic awareness and long-range temporal consistency. To address this limitation, we introduce a parameter-free refinement stage that recalibrates each raw score $a_u$ using cross-modal retrieval cues. Let $\mathbf{x}_u$ denote the visual embedding of segment $V_u$. The refinement of $a_u$ proceeds in three stages.

**1) Global semantic retrieval.** We first retrieve semantically related descriptions $\{y_k\}_{k=1}^M$ using FAISS-based nearest-neighbor search in the shared ImageBind embedding space. The cosine-based text similarity is defined as:

$$S_{\text{text}}(u, k) = \cos(\mathbf{x}_u, \mathbf{y}_k), \tag{7}$$

where $\mathbf{y}_k$ is the embedding of description $y_k$.

This step captures global semantic alignment, but does not account for temporal proximity within the video.

**2) Local visual similarity with temporal weighting.** To enforce temporal coherence, we compute cosine similarity between $\mathbf{x}_u$ and all visual embeddings $\{\mathbf{x}_t\}$ from the same video:

$$S_{\cos}(u, t) = \cos(\mathbf{x}_u, \mathbf{x}_t). \tag{8}$$

We modulate this similarity using a temporal weighting function:

$$w_t = \begin{cases} 1 + 0.5 \exp\left(-\dfrac{|t - u|}{2\Delta}\right), & |t - u| \leq W\Delta, \\ 1, & \text{otherwise}, \end{cases} \tag{9}$$

where $\Delta$ is the frame interval and $W$ is the temporal window size.

The temporally-weighted visual similarity becomes:

$$S_{\text{visual}}(u, t) = S_{\cos}(u, t) \cdot w_t. \tag{10}$$

This emphasizes visually similar segments that are also temporally close to $V_u$, improving local stability.

**3) Cross-modal re-ranking.** Each retrieved descriptiopn $y_k$ is associated with a set of video frames $\mathcal{F}(k)$. For candidate $k$, we compute the maximal visual similarity:

$$\hat{S}_{\text{visual}}(u,k) = \max_{t \in \mathcal{F}(k)} S_{\text{visual}}(u,t). \tag{11}$$

We then fuse text and visual similarities:

$$S(u,k) = \alpha S_{\text{text}}(u,k) + (1-\alpha)\hat{S}_{\text{visual}}(u,k), \quad \alpha \in [0,1]. \tag{12}$$

The top-$K$ candidates under $S(u,k)$ form the refined neighbor set $\mathcal{K}_u$.

**4) Similarity-weighted score aggregation.** The refined anomaly score for segment $u$ is obtained by softmax-weighted aggregation of the raw scores from its re-ranked neighbors:

$$\hat{a}_u = \sum_{k \in \mathcal{K}_u} a_k \frac{\exp(S(u,k))}{\sum_{k' \in \mathcal{K}_u} \exp(S(u,k'))}. \tag{13}$$

By jointly integrating global semantic alignment (Eq. 7) and temporally-aware visual consistency (Eqs. 9–10), the refinement stage produces smoother, context-aware anomaly scores without introducing additional trainable parameters.

## 4 Experimental Setup

In this section, we systematically evaluate Cog-VADU to validate its effectiveness from three complementary perspectives.

**(Q1) Detection Effectiveness.** Does sequential cognitive prompting with contextual refinement improve zero-shot video anomaly detection? We evaluate detection performance on large-scale surveillance benchmarks.

**(Q2) Reasoning Depth.** Does structured anomaly prompting enable deeper semantic understanding beyond binary detection? We assess reasoning capability on dedicated anomaly-understanding benchmarks.

**(Q3) Generalization.** How robust is Cog-VADU across datasets with diverse anomaly distributions, annotation protocols, and evaluation criteria?

**Detection Benchmarks.** **UCF-Crime** Sultani et al. (2019) contains 13 anomaly categories and 290 test videos (140 abnormal), with an average duration of 2.13 minutes. **XD Violence** Wu et al. (2020) includes six anomaly types and 800 test videos (500 abnormal), with an average duration of 1.62 minutes.

**Evaluation Metrics.** Following prior zero-shot VAD works Zanella et al. (2024); Ye et al. (2025), We report frame-level Area Under the Curve (AUC). For XD-Violence, we additionally report Average Precision (AP).

**Reasoning Benchmarks. VANE-Bench** Bharadwaj et al. (2025) contains 325 clips and 559 multiple-choice QA pairs across nine sub-datasets, designed to evaluate semantic anomaly understanding. **HIVAU-70k** Zhang et al. (2025b) evaluates hierarchical anomaly reasoning at clip-, event-, and video-level granularity. Annotations are constructed via semi-automated segmentation and recursive description.

**Evaluation Metrics.** To evaluate semantic anomaly understanding, performance on VANE-Bench Bharadwaj et al. (2025), we measured MCQ accuracy . For HIVAU-70k Zhang et al. (2025b), we report BLEU, METEOR, and ROUGE-L. Together, these benchmarks evaluate detection accuracy, structured reasoning ability, and cross-dataset generalization under a fully training-free setting.

### 4.1 Benchmarks and Unified Re-annotation

**Unified Re-annotation of Cause-Effect Relationships.** While evaluating on UCF-Crime Sultani et al. (2019) and XD-Violence Wu et al. (2020), we identified systematic annotation inconsistencies that penalize reasoning-centric models. Specifically, we observed a "trigger-bias" where only the initial cause of an anomaly (e.g., a gunshot) is labeled, while semantically anomalous effects (e.g., victims collapsing, smoke, or aftermath) are omitted. Additionally, repetitive anomalies are often inconsistently marked, frequently appearing at $t_n$ without labels despite a labeled occurrence at $t_1$. As illustrated in Fig. 5, these gaps create a structural mismatch between visual reasoning and ground-truth labels. To address this, we re-annotated both datasets using a unified *causal-chain policy* that captures the complete temporal extent of each event-from the initiating cause to the resulting effect and escalation. This protocol increased anomalous frame counts by 39.1% and 22.5% for UCF-Crime and XD-Violence, respectively. *Detailed statistics and the full annotation protocol are provided in the Supplementary Material (Sec:B).*

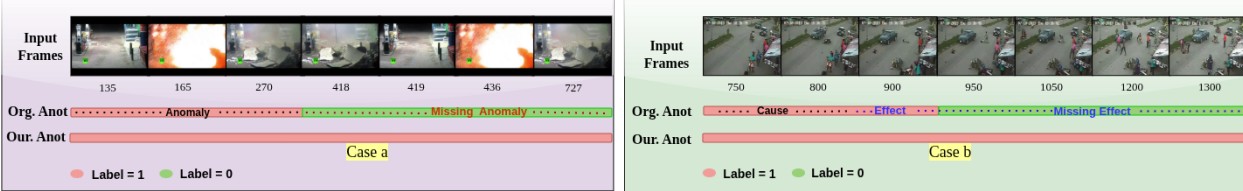

Figure 5: **UCF Sultani et al. (2019) Re-Annotation Examples.** Case (a) shows missing anomaly regions in the original annotations. Case (b) highlights cause–and–effect anomaly segments previously ignored but included in our unified annotations.

**Implementation of Cog-VADU.** We adopt *VideoLLaMA3-7B* Zhang et al. (2025a) as the default backbone $f_{\text{LVLM}}$ for Cog-VADU, and report all results under this setting unless otherwise specified. For ablation studies, we additionally evaluate alternative backbones, including *VideoLLaMA3-2B* Zhang et al. (2025a) and *InternVL2-8B* Chen et al. (2024a). During the refinement stage, we use *ImageBind* Girdhar et al. (2023) to extract cross-modal retrieval features. We select $M = 30$ top visual–textual candidates, followed by softmax-weighted aggregation over the top $k = 10$ reranked candidates. In the raw stage, we sample $N = 10$ frames per temporal segment of length $T = 10$ seconds, with a frame interval $\Delta = 16$.

### Quantitative Results

We compare Cog-VADU with four representative VAD paradigms: (1) weakly supervised, (2) one-class unsupervised, (3) training-based explainable, and (4) training-free explainable methods. Results are reported in Table 2 and Table 3.

**Detection Performance.** Despite being fully training-free, Cog-VADU achieves a competitive AUC-ROC across both datasets. On XD-Violence, it reaches 90.15% AUC, surpassing the previous training-free state-of-the-art, LAVAD Zanella et al. (2024) (85.36%), and outperforming several training-based explainable approaches. This demonstrates that structured anomaly reasoning alone enables strong temporal discrimination without dataset-specific optimization.

**AP under Original Labels.** Although AUC remains high, Average Precision (AP) under the original annotations is lower than that of supervised models. As discussed in Section 4.1, this is due to the *trigger bias*, where only initiating causes are labeled while downstream effects are omitted. Supervised models implicitly learn to ignore these regions, whereas Cog-VADU assigns high anomaly scores to both causes and consequences. Consequently, correct causal reasoning may be penalized as false positives under an incomplete ground truth.

**Performance under Unified Re-annotation.** Under our unified re-annotation protocol ("R."), the precision gap largely disappears. This recovery confirms that Cog-VADU captures coherent cause–and–effect chains rather than isolated triggers and motion patterns, thus aligning more closely with human semantic understanding of anomalies.

Table 2: **AUC and AP (%) on XD-Violence** Wu et al. (2020). "O." and "R." denote original and re-annotated splits. "—" indicates codes/results unavailable.

| Method | AUC (O.) | AUC (R.) | AP (O.) | AP (R.) |
|---|---|---|---|---|
| **Weakly Supervised (Non-Explainable)** | | | | |
| RareAnom Thakare et al. (2023) | 68.33 | — | — | — |
| RTFM Tian et al. (2021) | — | — | 77.81 | — |
| MGFN Chen et al. (2022) | — | — | 80.11 | — |
| UR-DMU Zhou et al. (2023a) | **94.02** | **93.74** | 81.66 | 84.50 |
| **One-Class (Non-Explainable)** | | | | |
| BODS Wang & Cherian (2019) | 57.32 | — | — | — |
| GODS Wang & Cherian (2019) | 61.56 | — | — | — |
| **Training-Based (Explainable)** | | | | |
| VERA Ye et al. (2025) | 88.26 | 92.15 | 70.54 | 80.62 |
| Holmes-VAU Zhang et al. (2025b) | — | — | **87.68** | — |
| **Training-Free (Explainable)** | | | | |
| ZS CLIP Zanella et al. (2024) | 38.21 | — | 17.83 | — |
| ZS IB (Img) Girdhar et al. (2023) | 27.25 | — | — | — |
| ZS IB (Vid) Girdhar et al. (2023) | 25.36 | — | — | — |
| LLAVA-1.5 Liu et al. (2024) | 79.62 | — | 50.26 | — |
| LAVAD Zanella et al. (2024) | 85.36 | 88.72 | 62.01 | 75.44 |
| **Ours** | **90.15** | **94.32** | **65.15** | **95.24** |

Table 3: **AUC (%) on UCF-Crime** Sultani et al. (2019). "O." and "R." denote original and re-annotated splits.

| Method | AUC (O.) | AUC (R.) |
|---|---|---|
| **Weakly Supervised (Non-Explainable)** | | |
| CLAWS Zaheer et al. (2021) | 83.03 | — |
| RTFM Tian et al. (2021) | 84.03 | — |
| UR-DMU Zhou et al. (2023a) | 86.97 | 89.57 |
| MGFN Chen et al. (2022) | 86.98 | — |
| SSRL Li et al. (2022) | 87.43 | — |
| CLIP-TSA Joo et al. (2023) | **87.58** | — |
| **One-Class (Non-Explainable)** | | |
| BODS Wang & Cherian (2019) | 68.26 | — |
| GODS Wang & Cherian (2019) | **70.46** | — |
| **Training-Based (Explainable)** | | |
| VERA Ye et al. (2025) | 86.55 | 88.34 |
| Holmes-VAU Zhang et al. (2025b) | **88.96** | **89.52** |
| **Training-Free (Explainable)** | | |
| ZS CLIP Zanella et al. (2024) | 53.16 | — |
| ZS IB (Img) Girdhar et al. (2023) | 53.65 | — |
| ZS IB (Vid) Girdhar et al. (2023) | 55.78 | — |
| LLAVA-1.5 Liu et al. (2024) | 72.84 | — |
| LAVAD Zanella et al. (2024) | 80.28 | 83.80 |
| **Ours** | **80.54** | **86.06** |

## 4.2 Anomaly Understanding & Reasoning (Q2): Model-Agnostic Evaluation

We evaluate hierarchical anomaly reasoning on HIVAU-70K across clip-level (C), event-level (E), and video-level (V) granularities. Table 4 compares three groups: (1) training + instruction-tuned VAD MLLMs, (2) open-source MLLMs, and (3) open-source MLLMs augmented with CoADTP. Cog-VADU represents our full training-free framework.

**Baseline MLLMs Lack Structured Anomaly Reasoning.** General open-source MLLMs (e.g., Video-ChatGPT, Video-LLaVA, LLaVA-Next-Video, InternVL2) exhibit weak event- and video-level performance. Although some models achieve moderate clip-level BLEU, their scores collapse at longer temporal scales, indicating insufficient anomaly grounding and poor hierarchical reasoning.

**CoADTP is Model-Agnostic and Consistently Improves Anomaly Reasoning.** Adding CoADTP to the same backbones yields dramatic gains across all granularities. These consistent improvements across diverse architectures demonstrate that CoADTP is model-agnostic: it enhances anomaly reasoning without modifying model weights or requiring task-specific training. Structured chain-of-anomaly prompting alone substantially strengthens temporal coherence and causal understanding.

**Comparison with Training-Based VAD MLLMs.** Training-based models such as Hawk Tang et al. (2024), Holmes-VAD, Zhang et al. (2024b), and Holmes-VAU Zhang et al. (2025b) benefit from anomaly-focused supervision and hierarchical instruction tuning, with Holmes-VAU achieving the strongest clip-level BLEU and ROUGE-L scores due to dataset-specific optimization. In contrast, enforcing cognitive reasoning through CoADTP yields consistent improvements across multiple backbones, demonstrating that CoADTP is a robust, model-agnostic framework rather than a backbone-dependent enhancement.

**Beyond n-gram metrics: semantic faithfulness via BERTScore.** The metrics in Table 4 quantify lexical overlap with reference rationales, which can penalise valid paraphrases and reward surface-level word matches. To complement the n-gram evaluation, we additionally report BERTScore Zhang et al. (2020), which measures semantic similarity via contextual embeddings of RoBERTa-large Zhang et al. (2020) with baseline-rescaling. Whereas Table 4 partitions results by temporal granularity (clip, event, video), Table 5 partitions by HIVAU-70k task type (judgement, description, analysis, captioning), giving a complementary view of rationale quality across the four reasoning instructions the benchmark prescribes. We report F1 on the intersection of 350 IDs covered by every evaluated method.

Table 4: Comparison of hierarchical video anomaly reasoning performance across clip-level (C), event-level (E), and video-level (V) granularity. BLEU denotes cumulative BLEU-1 to BLEU-4 scores. Results demonstrate the effectiveness of reasoning-driven approaches for structured anomaly understanding from short clips to long videos. **Bold** means best and Underline means second best in **each group**. Results with + CoADTP show the impact of the Chain of Anomaly Detection Thought Prompt.

| Method | BLEU ↑ | | | METEOR ↑ | | | ROUGE-L ↑ | | |
|---|---|---|---|---|---|---|---|---|---|
| | C | E | V | C | E | V | C | E | V |
| **Training + Instruction-Tuned VAD MLLMs** | | | | | | | | | |
| Hawk Tang et al. (2024) | 0.320 | 0.165 | 0.202 | **0.228** | 0.191 | 0.196 | 0.156 | 0.104 | 0.114 |
| Holmes-VAD Zhang et al. (2024b) | *0.514* | *0.318* | *0.306* | *0.224* | **0.237** | **0.211** | *0.235* | *0.164* | *0.161* |
| Holmes-VAU Zhang et al. (2025b) | **0.913** | **0.804** | **0.566** | 0.190 | 0.165 | 0.121 | **0.329** | **0.370** | **0.355** |
| **Open-source MLLMs** | | | | | | | | | |
| Video-ChatGPT Maaz et al. (2024) | 0.152 | 0.068 | 0.066 | 0.102 | 0.069 | 0.044 | 0.153 | 0.048 | 0.079 |
| Video-ChatGPT + CoADTP(Ours) | 0.349 | 0.439 | 0.447 | 0.197 | 0.253 | 0.242 | 0.162 | 0.222 | 0.234 |
| Video-LLaVA Lin et al. (2024) | 0.164 | 0.046 | 0.055 | 0.097 | 0.022 | 0.014 | 0.132 | 0.023 | 0.045 |
| Video-LLaVA + CoADTP(Ours) | 0.391 | 0.213 | 0.230 | 0.256 | 0.209 | 0.199 | 0.183 | 0.131 | 0.137 |
| LLaVA-Next-Video Zhang et al. (2024d) | 0.435 | 0.091 | 0.120 | 0.117 | 0.085 | 0.096 | 0.198 | 0.080 | 0.106 |
| LLaVA-Next-Video + CoADTP(Ours) | 0.651 | **0.839** | **0.790** | 0.287 | **0.339** | 0.299 | 0.253 | 0.352 | 0.325 |
| QwenVL2 Wang et al. (2024) | 0.312 | 0.082 | 0.155 | 0.133 | 0.092 | 0.112 | 0.163 | 0.081 | 0.137 |
| QwenVL2 + CoADTP(Ours) | 0.470 | 0.792 | 0.722 | 0.215 | 0.307 | 0.281 | 0.237 | 0.341 | 0.323 |
| InternVL2 Chen et al. (2024b) | 0.331 | 0.101 | 0.145 | 0.141 | 0.095 | 0.101 | 0.182 | 0.102 | 0.122 |
| InternVL2 + CoADTP(Ours) | 0.550 | 0.830 | 0.766 | 0.247 | 0.323 | **0.299** | **0.281** | **0.357** | **0.342** |
| Video-R1 Feng et al. (2025) | 0.572 | 0.171 | 0.229 | 0.186 | 0.159 | 0.156 | 0.187 | 0.103 | 0.116 |
| Video-R1 + CoADTP(Ours) | 0.439 | 0.769 | 0.732 | 0.338 | 0.301 | 0.280 | 0.236 | 0.347 | 0.325 |
| Cog-VADU (Ours) | **0.785** | 0.739 | 0.743 | **0.349** | 0.297 | 0.290 | 0.278 | 0.329 | 0.324 |

Table 5: BERTScore F1 Zhang et al. (2020)on HIVAU-70k Zhang et al. (2025b) (intersection of 350 IDs covered by all 10 methods; RoBERTa-large Zhang et al. (2020), baseline-rescaled). Cog-VADU achieves training-free performance competitive with strong CoADTP-augmented backbones while substantially outperforming the training-based explainable baseline Holmes-VAD Zhang et al. (2024b). Best in each column is in **bold**; second-best is underlined.

| Method | Judg. | Desc. | Anal. | Capt. | Overall |
|---|---|---|---|---|---|
| *Training + instruction-tuned VAD MLLMs* | | | | | |
| HAWK Tang et al. (2024) | 0.078 | 0.075 | 0.071 | 0.077 | 0.077 |
| Holmes-VAD Zhang et al. (2024b) | 0.167 | 0.236 | 0.235 | 0.263 | 0.245 |
| Holmes-VAU Zhang et al. (2025b) (SOTA, in-domain) | **0.652** | **0.297** | **0.479** | **0.430** | **0.431** |
| *Open-source MLLMs + CoADTP (zero-shot)* | | | | | |
| Video-ChatGPT Maaz et al. (2024) + CoADTP | 0.400 | 0.170 | 0.084 | 0.104 | 0.149 |
| Video-LLaVA Lin et al. (2024) + CoADTP | 0.114 | 0.081 | 0.129 | 0.143 | 0.127 |
| LLaVA-Next-Video Zhang et al. (2024d) + CoADTP | 0.555 | 0.197 | 0.243 | 0.333 | 0.326 |
| QwenVL2 Wang et al. (2024) + CoADTP | 0.507 | 0.237 | 0.259 | 0.333 | 0.330 |
| InternVL2 Chen et al. (2024b) + CoADTP | 0.562 | 0.235 | 0.277 | 0.355 | 0.350 |
| Video-R1 Feng et al. (2025) + CoADTP | 0.516 | 0.224 | 0.288 | 0.269 | 0.288 |
| **Cog-VADU (Ours, VideoLLaMA3 Zhang et al. (2025a))** | 0.545 | 0.223 | 0.264 | 0.309 | 0.316 |

**Key findings.** Four observations follow from Table 5. *First*, Cog-VADU attains an overall F1 of 0.316 *without any task-specific training*, substantially outperforming the training-based explainable baseline Holmes-VAD Zhang et al. (2024b) (0.245) and HAWK Tang et al. (2024) (0.077). *Second*, on the *judgement* task — which directly tests the model's anomaly decision — Cog-VADU achieves F1 = 0.545, the second-highest among all methods and close to Holmes-VAU's 0.652 despite never being trained on HIVAU-70k. *Third*, the gains from CoADTP scale with backbone capability: stronger backbones (InternVL2, QwenVL2, LLaVA-Next-Video, Video-R1) cluster in the 0.29–0.35 range, while older backbones with weaker instruction-

following (Video-ChatGPT, Video-LLaVA) remain at 0.13–0.15. This pattern — consistent with the cross-backbone trends in Table 4 — indicates that CoADTP *amplifies the reasoning capacity already present in the underlying LVLM* rather than substituting for it. *Fourth,* the gap to Holmes-VAU Zhang et al. (2025b) (0.431) is attributable to its in-domain instruction tuning on HIVAU-70k; given that Cog-VADU is fully training-free and incurs no annotation cost, this trade-off is expected and acceptable.

Taken together, BERTScore confirms that Cog-VADU's predictions are *semantically faithful* to the references even when phrasing differs from the ground truth, complementing rather than replacing the n-gram metrics in Table 4.

### 4.3 Performance Comparison on VANE-Bench & Generalization (Q3)

As shown in Table 6, we evaluate our zero-shot, reasoning-based VAD framework Cog-VADU on VANE-Bench Bharadwaj et al. (2025), a challenging benchmark designed to assess video anomaly understanding through multi-type, multiple-choice questions. Results are reported in terms of anomaly detection accuracy. Despite not relying on any task-specific training or instruction tuning, our **Cog-VADU** achieves competitive performance against strong baselines, demonstrating the effectiveness of structured cognitive reasoning for anomaly detection.

Table 6: **Performance comparison on VANE-Bench.** Accuracy (%) across different video generation models and surveillance datasets. **bold** means best and underline means second best in a group.

| Method | SORA | OpenSORA | RG2 | VideoLCM | MS-T2 | Avenue | Ped1 | Ped2 |
|---|---|---|---|---|---|---|---|---|
| **MLLM-based VAD (Training+ instruction Tuned )** | | | | | | | | |
| Holmes-VAU Zhang et al. (2025b) | 2.17 | 34.00 | 24.00 | 29.81 | 25.00 | 6.06 | 3.33 | 5.56 |
| Holmes-VAD Zhang et al. (2024b) | 6.52 | 34.00 | 32.00 | 33.56 | 22.92 | 12.12 | 20.00 | 5.56 |
| HAWK Tang et al. (2024) | 24.64 | 52.00 | 44.00 | 36.54 | 50.00 | 36.36 | 36.67 | 38.89 |
| VAD-R1 Huang et al. (2025) | **41.30** | **78.00** | **56.00** | **63.46** | **60.42** | **75.76** | **60.00** | **63.89** |
| **Open-Source MLLMs** | | | | | | | | |
| Video-LLaMA Zhang et al. (2023) | 11.59 | 18.00 | 16.00 | 10.57 | 10.41 | 30.00 | 16.66 | 5.55 |
| VideoChat Li et al. (2024) | 10.74 | 28.00 | 4.00 | 17.64 | 20.83 | 32.25 | 13.33 | 13.88 |
| Video-ChatGPT Maaz et al. (2024) | **26.47** | 22.00 | 12.00 | 18.26 | 16.66 | 39.39 | 40.00 | 19.44 |
| Video-LLaVA Lin et al. (2024) | 10.86 | 18.00 | 16.00 | 19.23 | 16.66 | 3.03 | 2.77 | 6.06 |
| MovieChat Song et al. (2024) | 8.69 | 10.00 | 16.00 | 14.42 | 6.25 | 18.18 | 6.66 | 11.11 |
| LLaMA-VID Li et al. (2023b) | 7.97 | 14.00 | 20.00 | 19.23 | 14.58 | 27.27 | 6.66 | 19.44 |
| TimeChat Ren et al. (2024) | 21.73 | 26.00 | 28.00 | 22.11 | 20.83 | 24.20 | 27.58 | 11.11 |
| **Cog-VADU(Reasoning-Based Training-Free Ours)** | 24.64 | **80.00** | **60.00** | **50.00** | **50.00** | 72.73 | **56.67** | **72.22** |

Importantly, while most existing methods depend on supervised fine-tuning or instruction-tuned representations, Cog-VADU operates in a purely training-free, zero-shot setting. This highlights its strong generalization capability, as it can transfer to diverse anomaly scenarios without exposure to domain-specific annotations. The results suggest that explicitly modeling reasoning steps, rather than relying solely on learned representations, can serve as a powerful alternative for video anomaly understanding.

**Qualitative comparison on VANE.** Fig. 6 further present qualitative examples from VANE-Bench. Our method generates structured and interpretable outputs that explicitly connect *scene understanding → abnormal event localization → causal reasoning → final decision.* Compared to baseline methods, which often produce shallow or implicit predictions, Cog-VADU provides grounded, step-by-step explanations that enhance both interpretability and reliability. This demonstrates that our approach not only predicts anomalies effectively but also explains *why* they occur, aligning with the goal of trustworthy video understanding systems.

### 4.4 Ablation Studies

We conduct comprehensive ablation studies on the UCF-Crime dataset Sultani et al. (2019) to systematically analyze the contribution of each component in our framework. In particular, we investigate how different design choices affect the quality, stability, and reliability of anomaly predictions.

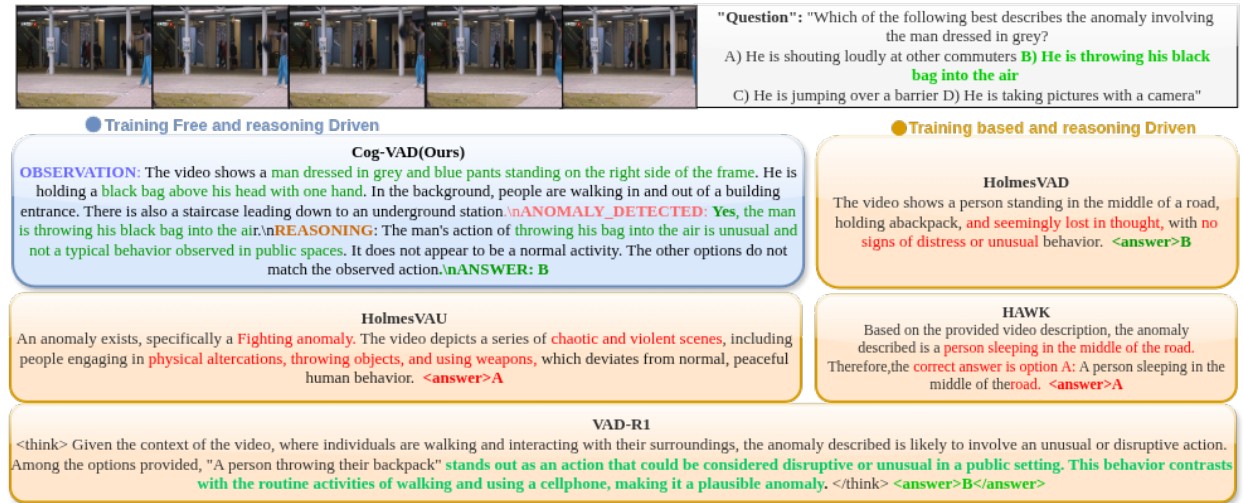

Figure 6: **Qualitative comparison on VANE.** Cog-VADU generates structured, evidence-grounded reasoning (Observation → Anomaly → Reasoning → Answer), correctly identifying the anomaly (B). In contrast, prior multimodal VAD methods, including RL-based VAD-R1 Bharadwaj et al. (2025), often produce weakly grounded explanations.

Specifically, we examine: (1) alternative refinement strategies to understand the role of cross-modal alignment, (2) the impact of top-$K$ semantic neighbor selection on contextual aggregation, (3) the influence of the visual-textual fusion weight $\alpha$ on balancing global semantic and local temporal cues, and (4) the choice of vision-language backbone $f_{\text{LVLM}}$ to evaluate model generalization.

Through these analyses, we aim to answer three key questions: *(i)* how much each component contributes to performance gains, *(ii)* how sensitive the framework is to parameter variations, and *(iii)* whether the proposed design leads to stable and consistent predictions across diverse settings.

**Effect of refinement strategies.** Table 7 compares different refinement mechanisms. Our *Cross-Modal Re-Ranking* consistently outperforms both visual-textual and visual-visual alternatives, achieving the highest AUC and AP.

Visual-visual matching primarily captures local temporal consistency, but often fails to incorporate global semantic context. In contrast, visual-textual matching emphasizes global semantic alignment, while lacking sensitivity to local temporal coherence. Our approach integrates both perspectives, jointly modeling global semantics and local temporal relationships, leading to more reliable and robust anomaly scoring.

| Strategy | AUC | AP |
|---|---|---|
| Visual–Textual Zanella et al. (2024) | 0.722 | 0.211 |
| Visual–Visual Ye et al. (2025) | 0.755 | 0.249 |
| **Ours (Cross-Modal Re-Ranking)** | **0.805** | **0.381** |

Table 7: **Ablation: Refinement Strategies (UCF-Crime Sultani et al. (2019)).** Frame-level AUC and AP for different strategies.

**Effect of Refinement Parameters.** We further analyze the impact of key parameters that govern the refinement stage, including the number of retrieved candidates ($M$), the temporal aggregation window ($K$), and the fusion weights ($\alpha_{\text{vis}}, \alpha_{\text{text}}$). These parameters control semantic retrieval, temporal smoothing, and cross-modal alignment, which are critical for producing stable and reliable anomaly predictions. As shown in Table 8, increasing $M$ improves performance

**Analysis.**

- **Top-$M$:** Increasing the number of retrieved semantic candidates improves performance by providing richer contextual evidence. Small $M$ leads to unstable alignment, while larger $M$ captures diverse context. However, very large $M$ may introduce noise. Empirically, $M = 30$ achieves the best trade-off.

- **Temporal window $K$:** Larger $K$ improves temporal consistency by smoothing predictions and capturing cause-and-effect relations. However, excessive smoothing may blur short anomalies. Optimal performance is observed for $K \in [8, 10]$.

- **Fusion weights:** The fusion weights balance the contribution of visual similarity and semantic alignment. Visual similarity captures local temporal coherence, while textual similarity provides global semantic context. An imbalanced weighting biases the model toward either local or global cues, leading to suboptimal performance. In contrast, balanced fusion ($\alpha_{\text{vis}} = \alpha_{\text{text}} = 0.5$) effectively integrates both signals, resulting in stable and context-aware anomaly predictions.

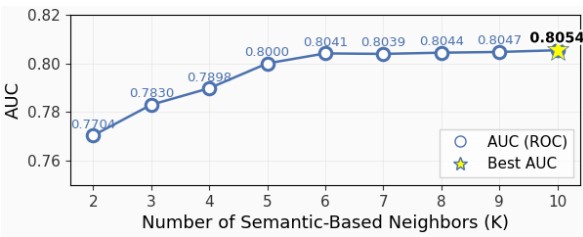

| **Top-$M$** | $\alpha_{\text{vis}}$ | $\alpha_{\text{text}}$ | **AUC** | **AP** |
|---|---|---|---|---|
| 10 | 0.5 | 0.5 | 0.7635 | 0.2907 |
| 20 | 0.5 | 0.5 | 0.7994 | 0.3416 |
| 25 | 0.5 | 0.5 | 0.8018 | 0.3636 |
| 30 | 0.5 | 0.5 | **0.8054** | **0.3808** |
| 50 | 0.5 | 0.5 | 0.8044 | 0.4012 |
| $K = 4$ | – | – | 0.7898 | 0.3347 |
| $K = 6$ | – | – | 0.8041 | 0.3632 |
| $K = 10$ | – | – | **0.8054** | **0.3808** |

Table 8: **Ablation of refinement parameters.** Influence of $M$, temporal window $K$, and fusion weights on AUC and AP.

Overall, these refinements harmonize semantic context, temporal consistency, and cross-modal alignment. *More ablations are provided in the Appendix C*

# 5 Concluding Remarks

We presented Cog-VADU, a training-free framework for video anomaly detection and understanding that models anomalies through sequential cognitive reasoning over video segments. By leveraging a model-agnostic chain-of-thought prompting strategy with CoADTP for step-wise rationale propagation and a cross-modal re-ranking stage, Cog-VADU produces temporally coherent and interpretable anomaly scores without requiring fine-tuning. Experiments on UCF-Crime, XD-Violence, VANE, and HIVAU-70k demonstrate competitive zero-shot performance and strong generalization, highlighting the effectiveness of structured cognitive reasoning and causal understanding for real-world anomaly detection. In addition, we re-annotated two major datasets in a unified manner by incorporating explicit cause-and-effect relationships, which are crucial for analyzing anomalies in real-world scenarios.

**Broader Impact Statement**

This work demonstrates how the reasoning capabilities of foundation models can enable training-free video anomaly detection with interpretable outputs. By decomposing complex scenarios into structured steps, the approach improves both performance and trust in AI systems for video surveillance, such as public safety. It also shows strong potential for generalization in open-world settings. However, like all video surveillance technologies, deployment must carefully consider privacy, fairness, and responsible use. We discuss below the position Cog-VADU takes on four dimensions that we view as central to responsible deployment: privacy, false positives and over-surveillance, bias and fairness, and broader responsible-deployment guidelines.

**Privacy.** Cog-VADU is training-free and uses a frozen LVLM. Unlike supervised VAD systems that require large, labelled surveillance corpora — which themselves raise consent, storage, and re-identification concerns, Cog-VADU introduces *no* additional surveillance data collection or storage requirement for model adaptation. To support privacy-respecting deployment, we recommend that operators (i) restrict inference to videos collected under documented consent or legal authorisation, (ii) avoid persistent storage of frame-level ratio-

nales that could be used for re-identification, and (iii) operate behind on-premise or institutionally-controlled inference rather than third-party APIs that may retain inputs.

**False Positives and Over-Surveillance.** Two properties of Cog-VADU are relevant to mitigating false alarms and over-surveillance. First, the framework produces a *calibrated continuous score together with a textual rationale* for each segment, rather than a binary alert. This makes the system amenable to human-in-the-loop review, since operators can inspect the rationale before acting on a positive prediction. Second, our wrestling case study (Appendix D, Fig. 12) shows that Cog-VADU correctly assigns a low score (0.15) to a visually high-motion-yet-benign scenario where motion-statistical methods (e.g., URDMU Zhou et al. (2023a)) trigger near-saturated false alarms ($\approx 0.999$). Semantic grounding therefore directly reduces the false-alarm rate driven by motion statistics alone. We recommend that any deployment using Cog-VADU adopts a human-review protocol for positive predictions, and we explicitly disclaim its use as an autonomous alerting system.

**Bias and Fairness.** Cog-VADU inherits the biases of its underlying LVLM (VideoLLaMA3-7B Zhang et al. (2025a)) and its anomaly database. While the training-free design avoids introducing additional dataset-specific bias from labelled surveillance corpora, it does not eliminate biases already present in the LVLM. The textual rationale, however, provides an *auditability handle* not available in non-explainable baselines: biased decisions can be inspected and flagged at the explanation level, rather than emerging from an opaque score. We recommend that practitioners audit Cog-VADU's outputs against demographic and contextual subgroups before any real-world deployment, and document any subgroup disparities as part of the deployment record.

**Responsible Deployment.** We state explicitly that:

- Cog-VADU is intended as a *decision-support tool for human operators*, not as an autonomous decision-making system.

- It should not be used for predictive policing, individual identification, or any setting where its outputs directly trigger consequential actions without human review.

- Known failure modes documented in this paper (e.g., the categorical labelling failure illustrated in Fig. 14) must be disclosed to end users so that operators are aware of the framework's limitations.

- Use in high-stakes contexts (transportation safety, public health emergencies) requires rigorous validation on the specific deployment distribution before any reliance is placed on its predictions.

In summary, Cog-VADU's training-free design and explainable outputs offer concrete advantages for privacy-respecting and auditable deployment, but these benefits depend on responsible deployment practices, human oversight, and recognition of the framework's documented failure modes.

## Acknowledgements

**Code and Data Availability.** The code and re-annotated datasets will be made publicly available upon acceptance.

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

# A  Appendix

This appendix provides supplementary information for the main paper. Firstly, we provide detailed information about the proposed unified re-annotated dataset, including the limitations of current annotations, the construction process, statistical analysis, and examples. Then, we present additional experimental details, including prompt design and model ablations. Finally, we provide extended results, visualizations, and discuss limitations and future directions.

# B  A Unified Annotation Protocol

## B.1  Motivation and Benchmark Limitations

Recent advances in video understanding Zhang et al. (2025d); Nadeem et al. (2025); Araghi et al. (2025) highlight the importance of modeling causal and temporal relationships, rather than relying solely on instantaneous visual cues. Benchmarks such as NExT-QA Xiao et al. (2021) and STAR Wu et al. (2024) explicitly evaluate cause-effect reasoning, emphasizing that meaningful video interpretation often depends on contextual and temporal dependencies. This perspective suggests that anomaly detection should extend beyond visually salient triggers (e.g., explosions) to include semantically meaningful states that follow such events.

Motivated by this, we propose a unified re-annotation protocol to better capture context-driven anomalies that are often overlooked in existing VAD benchmarks. Through analysis of XD-Violence Wu et al. (2020) and UCF-Crime Sultani et al. (2019), we identify key annotation limitations. Current annotations primarily focus on *cause-based* anomalies, while frequently omitting *effect-based* or context-dependent states, such as post-incident outcomes or prolonged abnormal situations.

These omissions introduce a structural bias. Training-based methods tend to ignore subtle or contextual anomalies to avoid false positives, whereas reasoning-based models detect them but are penalized under incomplete ground truth. This discrepancy contributes to reduced evaluation performance, particularly in metrics such as AP.

We further observe incomplete and inconsistent annotations, where repeated anomalies are labeled only once, and temporally dependent segments are partially or entirely missed. As illustrated in Fig. 7, repeated anomalous events are often annotated only at their first occurrence, while subsequent identical events remain unlabeled. Similarly, in Case b of Fig. 7, the causal event triggering the anomaly is annotated, whereas the subsequent effect segments, despite representing the same anomalous outcome, are omitted.

To address this, our unified protocol extends annotations to include these effect-based and temporally consistent anomaly segments. This results in more complete anomaly coverage and improved temporal consistency. Using this protocol, we construct refined annotations for both UCF-Crime Sultani et al. (2019) and XD-Violence Wu et al. (2020).

## B.2  Systematic Annotation Errors

Our frame-level analysis reveals three recurring issues:

**(1) Missed Effect Regions.** Only the initiating cause is labeled, while downstream consequences (e.g., injured victims, smoke-filled environments, evacuation scenes) remain unlabeled.

**(2) Repetition Bias.** Repeated anomalous actions are frequently annotated only once, ignoring later identical occurrences.

**(3) Inconsistent Cause-Effect Handling.** Videos containing causal chains (e.g., throwing an object $\rightarrow$ explosion $\rightarrow$ evacuation) are inconsistently labeled, sometimes marking only the cause, sometimes only the effect, and sometimes neither comprehensively.

These inconsistencies are substantially more prevalent in XD-Violence Wu et al. (2020), explaining the larger performance gain we observe after correction.

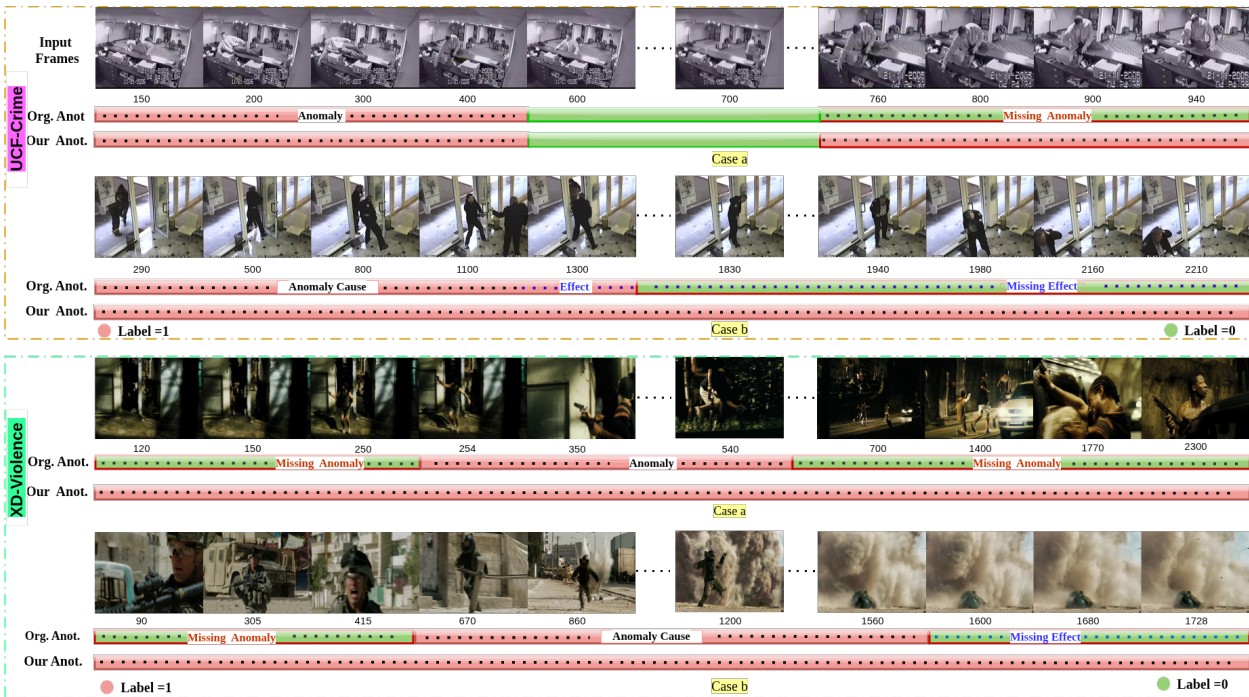

Figure 7: **UCF Sultani et al. (2019) and XD-Violence Wu et al. (2020) Re-Annotation Examples.** Case (a) shows missing anomaly regions in the original annotations but included in our unified annotations (**orig.Anot: Original Annotations, Our. Anot: Ours Re-Annotations**). Case (b) highlights cause–and–effect anomaly segments previously ignored but included in our unified annotations.

## B.3 Unified Annotation Principle

To address these inconsistencies and establish a reliable ground truth, we adopt a unified annotation principle:

> *All frames depicting the cause, effect, temporally connected escalation, or the aftermath of a real-world anomaly are labeled as anomalous.*

Under this protocol, both XD-Violence Wu et al. (2020) and UCF-Crime Sultani et al. (2019) are re-annotated to ensure complete cause-and-effect coverage, consistent labeling of repeated anomalous events, inclusion of escalation and post-incident aftermath, removal of partial temporal truncations, and annotation of subtle contextual anomalies that are not purely motion-driven. This correction increased anomalous frames in XD-Violence Wu et al. (2020) from 539,562 to 661,201 (+121,639 frames, **+22.5%**), and in UCF-Crime Sultani et al. (2019) from 88,263 to 122,757 (+34,494 frames, **+39.1%**).The statistical overview of the original vs ours unified reannotation has been shown in 8. All experiments in this paper are reported on both the original and unified annotations for fair comparison.

## B.4 Structural Analysis of Re-Annotations

To further characterize the structural differences between the original and unified labels, we present word-cloud visualizations of anomaly taxonomies and temporal-relation terms in Fig. 9. The anomaly perception word cloud highlights previously under-represented semantic states, while the temporal relation cloud emphasizes the necessity of modeling full cause-and-effect chains. Together, these analyses provide quantitative and linguistic evidence that the original annotations systematically under-represent semantically valid anomalous states.

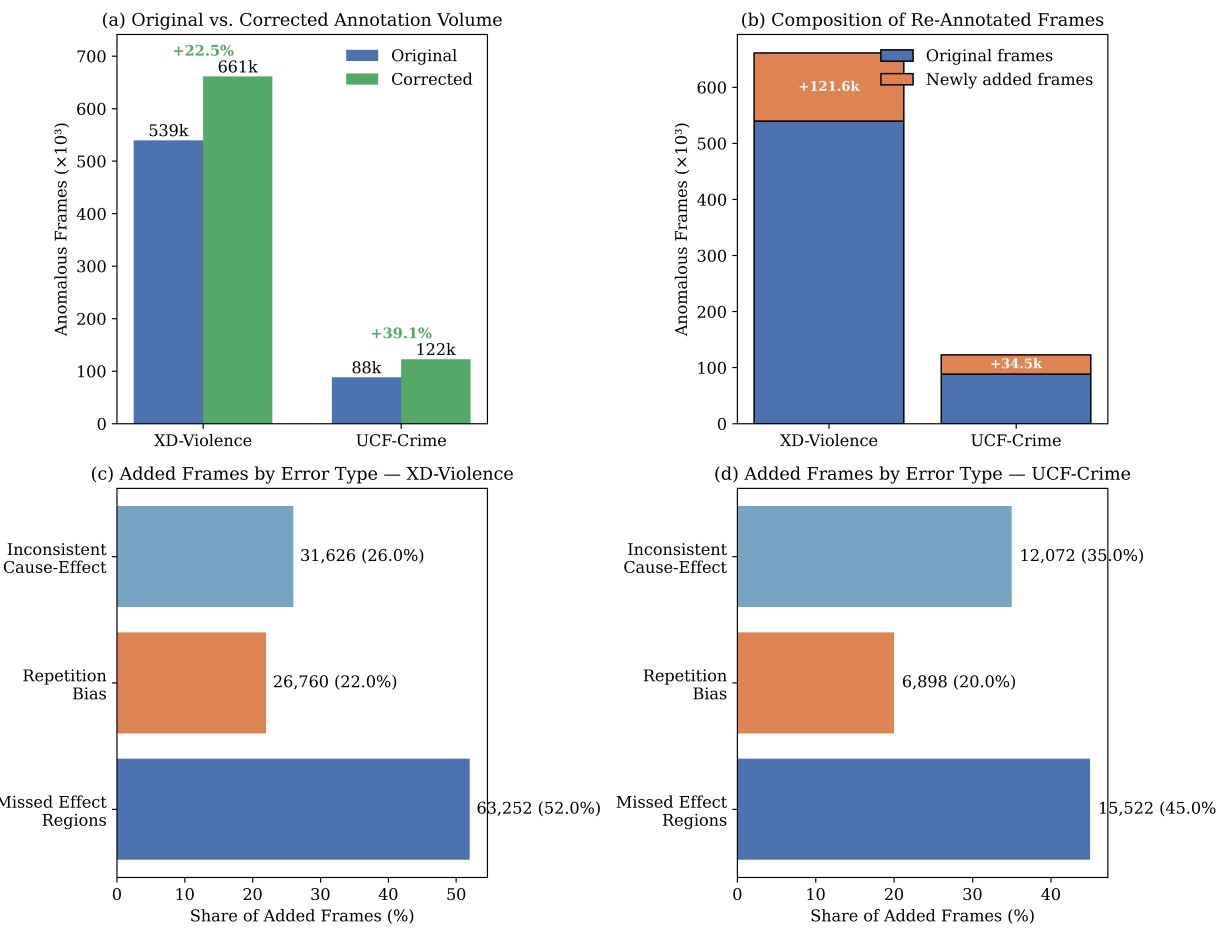

Figure 8: Quantitative analysis of re-annotation changes under the proposed unified labeling protocol. (a) Original vs. corrected anomalous frame volumes. (b) Composition of re-annotated frames showing original and newly added portions. (c–d) Distribution of added frames across three identified annotation error types for XD-Violence Wu et al. (2020) and UCF-Crime Sultani et al. (2019) , respectively.

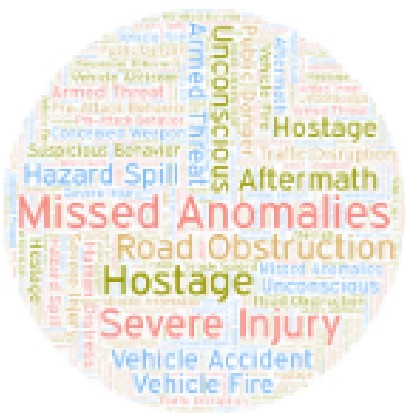 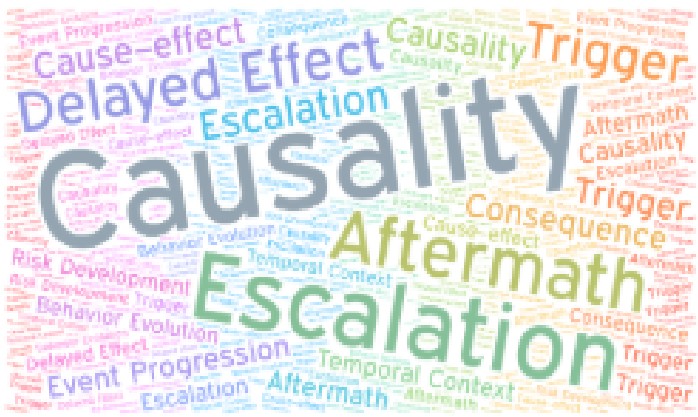

Figure 9: Structural analysis of annotation distributions under the proposed unified protocol. (a) Anomaly perception word cloud illustrating semantically enriched anomaly categories introduced through re-annotation, including subtle, contextual, and effect-based states that were under-represented in the original benchmarks. (b) Temporal relation word cloud highlighting causal, escalation, and aftermath relations, emphasizing the necessity of modeling complete cause-effect chains rather than isolated trigger events. Together, these visualizations provide linguistic and structural evidence that the original annotations are biased toward trigger-centric labeling, while our unified protocol captures temporally consistent and semantically grounded anomaly states.

## B.5 Annotation Procedure and Quality Control

The unified re-annotations described in Sections B–B.4

were produced through a *fully manual* process. No vision–language model, automatic relabeling system, or model-in-the-loop pseudo-labeling was used at any stage. This design choice is deliberate: because Cog-VADU is itself a VLM-based method, using any VLM to generate or assist the ground-truth labels would risk circular evaluation, where the labels are biased toward the same model family being evaluated. Manual annotation ensures that the corrected ground truth is *independent* of the models under test.

**Annotation protocol.** Annotators followed the fixed unified principle stated in Section B.3: all frames depicting the cause, effect, temporally connected escalation, or aftermath of a real-world anomaly are labeled as anomalous. To operationalise this principle consistently, each video was reviewed for three specific extensions to the original labels: (i) repeated anomalous events that were annotated only at first occurrence, (ii) effect-based anomaly continuations following a causal trigger, and (iii) semantically anomalous contextual states that are not purely motion-driven. Decisions were made on the basis of *temporal continuity and causal consistency* with the labeled event, rather than on visual saliency or model predictions.

**Verification and bias-reduction safeguards.** To reduce labeling errors and subjective bias, the annotations were subjected to the following safeguards:

- **Multiple inspection passes.** Each re-annotated video was reviewed in more than one pass, with the second pass specifically checking temporal boundaries (start/end frames of each extended segment) and the consistency of repeated-event labeling.

- **Protocol-driven decisions.** All corrections followed the fixed causal-chain protocol rather than ad-hoc judgement, ensuring that similar anomaly structures (e.g., cause → effect → aftermath) are labeled consistently across videos and across the two datasets.

- **Model-independence.** Because no model predictions were consulted during annotation, the corrected labels do not inherit biases from any particular detector or LVLM, avoiding circular reinforcement of model-specific behaviour.

- **Dual reporting.** To allow readers to assess the effect of re-annotation transparently, all experiments are reported on *both* the original and unified annotations (Sec. 4.1) Table 3 for Xd-Violence and Table 2 UCF crime datasets rather than replacing the original benchmark.

These safeguards are intended to ensure that the performance differences observed under the unified annotations reflect genuine improvements in anomaly coverage rather than artifacts of the re-labeling process.

## C  Chain-of-Anomaly Detection Thought Prompt (CoADTP)

The CoADTP is illustrated in Fig. 10. It is a step-wise prompting strategy that integrates three core components: (1) Chain of Thought style(Reasoning-based) (2) a textual anomaly knowledge base and (3) temporal feedback in the form of anomaly scores predicted and contextual information from previously processed clips for earlier segments. Incorporating these elements enables the model to perform temporally-aware reasoning, preserve contextual continuity across video segments, and refine anomaly predictions based on both accumulated evidence and domain-specific textual priors.

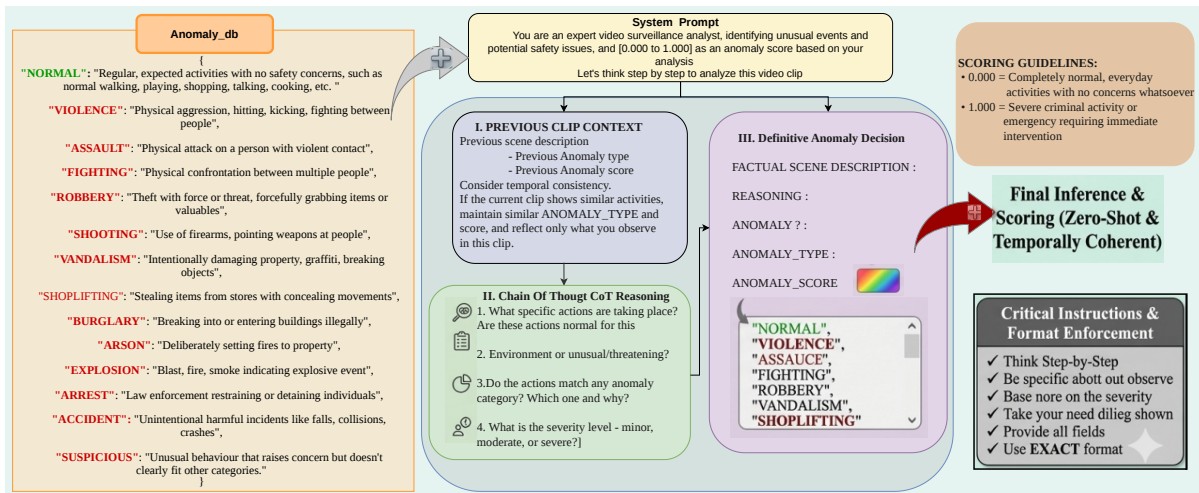

Figure 10: Workflow of CoADTP integrating anomaly database and temporal context and reasoning cues.

**Ablation on UCF-Crime** Sultani et al. (2019) is reported in Table 9 to evaluate the contribution of each Cog-VADU component. We present the ablation along two complementary axes: (i) *single-component contributions* (rows 2–4) where each textual component is added in isolation above the base prompt, and (ii) *cumulative configurations* (rows 5–7) that progressively build up to the full pipeline. In single-component rows, numerical score smoothing is disabled so that each component's effect is measured in isolation.

Compared to a basic prompt ("Describe this scene and generate an anomaly score between 0 and 1"), step-wise reasoning alone improves AUC from 0.7111 to 0.7548 (+4.37%) and AP from 0.2150 to 0.2488 (+3.38%), indicating more stable predictions. When evaluated in isolation, the Anomaly Database alone yields the largest standalone AP boost (AP 0.2150 → 0.3392, +12.42%) with a modest AUC gain (+2.95%), demonstrating that semantic priors are the strongest driver of anomaly *localization*. Temporal Feedback alone yields a comparable AP boost (+11.53%, AUC +3.28%), confirming the value of prompt-level temporal continuity even without numerical score smoothing. Each of the three textual components therefore *independently* improves over the base prompt — no component is redundant — and they contribute along different metric axes (Reasoning → AUC; Anomaly DB and Temporal FB → AP).

In cumulative configurations, incorporating the anomaly database on top of reasoning boosts AUC to 0.7810 (+2.62%) and AP to 0.3238. Adding temporal feedback further improves consistency across clips, yield-

ing AUC 0.7888 (+0.78%) and AP 0.3773 (+5.35%). Finally, the cross-modal re-ranking refinement stage achieves the best overall performance (AUC **0.8054**, AP **0.3808**), acting as a corrective post-processing mechanism that re-anchors textual rationales to visual evidence. Overall, the full Cog-VADU pipeline achieves a +9.43% AUC and +16.58% AP improvement over the base prompt baseline, with components contributing complementary rather than redundant gains.

Table 9: Ablation study on UCF-Crime showing the contribution of each Cog-VADU component. We report (i) the base prompt, (ii) each textual component evaluated in isolation above the base prompt (rows 2–4, with score smoothing disabled), and (iii) cumulative configurations leading to the full pipeline (rows 5–7). Reasoning is the strongest AUC driver; Anomaly DB and Temporal FB are the strongest AP drivers; components are complementary rather than redundant.

| Configuration | Reasoning | Anomaly Database | Temporal Feedback | Refinement | AUC | AP |
|---|---|---|---|---|---|---|
| Base Prompt | ✗ | ✗ | ✗ | ✗ | 0.7111 | 0.2150 |
| + Reasoning only | ✓ | ✗ | ✗ | ✗ | 0.7548 | 0.2488 |
| + Anomaly DB only | ✗ | ✓ | ✗ | ✗ | 0.7406 | 0.3392 |
| + Temporal FB only | ✗ | ✗ | ✓ | ✗ | 0.7439 | 0.3303 |
| + Reasoning + DB | ✓ | ✓ | ✗ | ✗ | 0.7810 | 0.3238 |
| + R + DB + Temporal FB | ✓ | ✓ | ✓ | ✗ | 0.7888 | 0.3773 |
| **Full Cog-VADU** | ✓ | ✓ | ✓ | ✓ | **0.8054** | **0.3808** |

## C.1 Choice of Large Vision Language Model

To assess the generalizability and model agnosticism of CoADTP across different LVLM architectures, we evaluate multiple model sizes and families (Table 10). We first test InternVL2-8B, a stronger variant within the InternVL2 series. We then examine the VideoLLaMA3 family, including its 2B variants, all running at `bfloat16` precision Zhang et al. (2025a). Our findings indicate that VideoLLaMA3 models demonstrate substantially improved temporal reasoning and scene understanding compared to InternVL2-8B, resulting in more stable anomaly scores, as illustrated in Figure 11. Notably, VideoLLaMA3-7B achieves the more promising performance, highlighting that improved cross-modal alignment and temporal comprehension contribute more than parameter count alone to anomaly detection effectiveness.

## C.2 Effect of Subsampling and Temporal Window

Following Ye et al. (2025), we adopt temporal-based uniform frame sampling for segment generation. Given a video $V$ with $F$ frames, we compute segment centers as

$$C = \{I_1,\ I_{d+1},\ I_{2d+1},\ \ldots,\ I_{(h-1)d+1}\},$$

where $d = 16$ is the interval between centers and $h = \lfloor F/d \rfloor$ is the number of segments. For each center frame $I_{(u-1)d+1}$, we define a temporal window of $T$ seconds around it and uniformly sample N frames to form the segment $S_u$.

We further study the effect of varying the window duration $T$ and sampling rate as shown in 11. Empirically, using a 10-second window with a sampling rate of 1 FPS yields the best performance, which aligns with the training FPS of VideoLLaMA3 Zhang et al. (2025a)

Table 10: AUC and AP (%) comparison across LVLM model sizes and model families.

| LVLM Model | AUC | AP |
|---|---|---|
| VideoLLaMA3-2B | 75.09 | 24.39 |
| InternVL2-8B | 76.26 | 31.01 |
| **VideoLLaMA3-7B** | **78.88** | **37.33** |

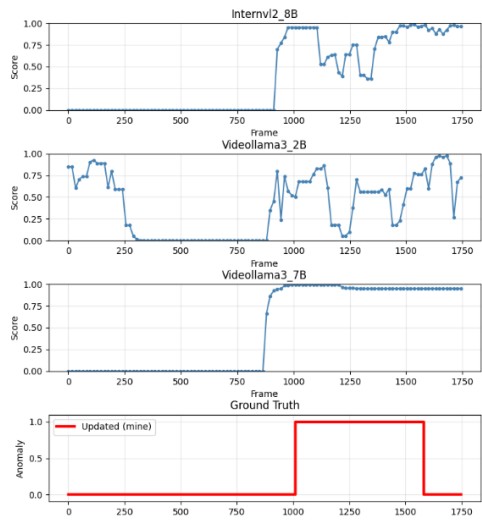

Table 11: Comparison of AUC and AP across different temporal window durations $T$ and number of sampled frames $N$.

| $T$ **(sec)** | $N$ | **AUC** | **AP** |
|---|---|---|---|
| 5 | 5 | 78.15 | 34.02 |
| 5 | 10 | 79.44 | 35.79 |
| 10 | 5 | 77.65 | 29.03 |
| 10 | 10 | **80.54** | **38.08** |

Figure 11: Output anomaly scores generated by InternVL2-8B Chen et al. (2024b), VideoLLaMA3-2B Zhang et al. (2025a), and VideoLLaMA3-7B Zhang et al. (2025a)

# D  Faithfulness and Failure Analysis

To substantiate the explainability claims made in the main paper, we conduct a focused qualitative study on three open-web videos that lie outside the training distribution of all evaluated baselines. These case studies serve three complementary purposes: *(i)* demonstrating that Cog-VADU's rationales are faithful to visual evidence; *(ii)* showing that the framework discriminates between visually similar but semantically different events; and *(iii)* documenting a failure mode of Cog-VADU and contrasting it with the failure modes of competing approaches.

## D.1  Symmetric Discriminative Cases

We deliberately select two videos that form a *symmetric pair*:

- **Case 1 — Visually anomalous, semantically normal:** A professional arm-wrestling match (John Cena vs. Mark Henry) that contains all surface cues commonly associated with anomalies (physical struggle, audience reaction, post-match injury), yet is contextually a sports event.

- **Case 2 — Visually subtle, semantically anomalous:** A nighttime recording of a missile interception over Tel Aviv, which is visually ambiguous (bright lights against a city skyline that could resemble fireworks) but corresponds to a genuine safety-critical event.

Together, these cases establish that Cog-VADU's reasoning is genuinely *context-driven* rather than biased toward a default visual pattern.

## D.2  Faithfulness of Cog-VADU Rationales

In both cases, Cog-VADU produces grounded, evidence-aligned rationales that connect observation → reasoning → decision. For Case 1, Cog-VADU correctly identifies the wrestling-entertainment context ("arm wrestling is a common activity in professional wrestling entertainment") and assigns a calibrated low score (**0.15**, NORMAL). For Case 2, Cog-VADU recognises the rapid trajectories as "possibly projectiles or explosions" and assigns a high score (**0.95**, EXPLOSION). The rationales are factually consistent with the visual evidence and aligned with the final score.

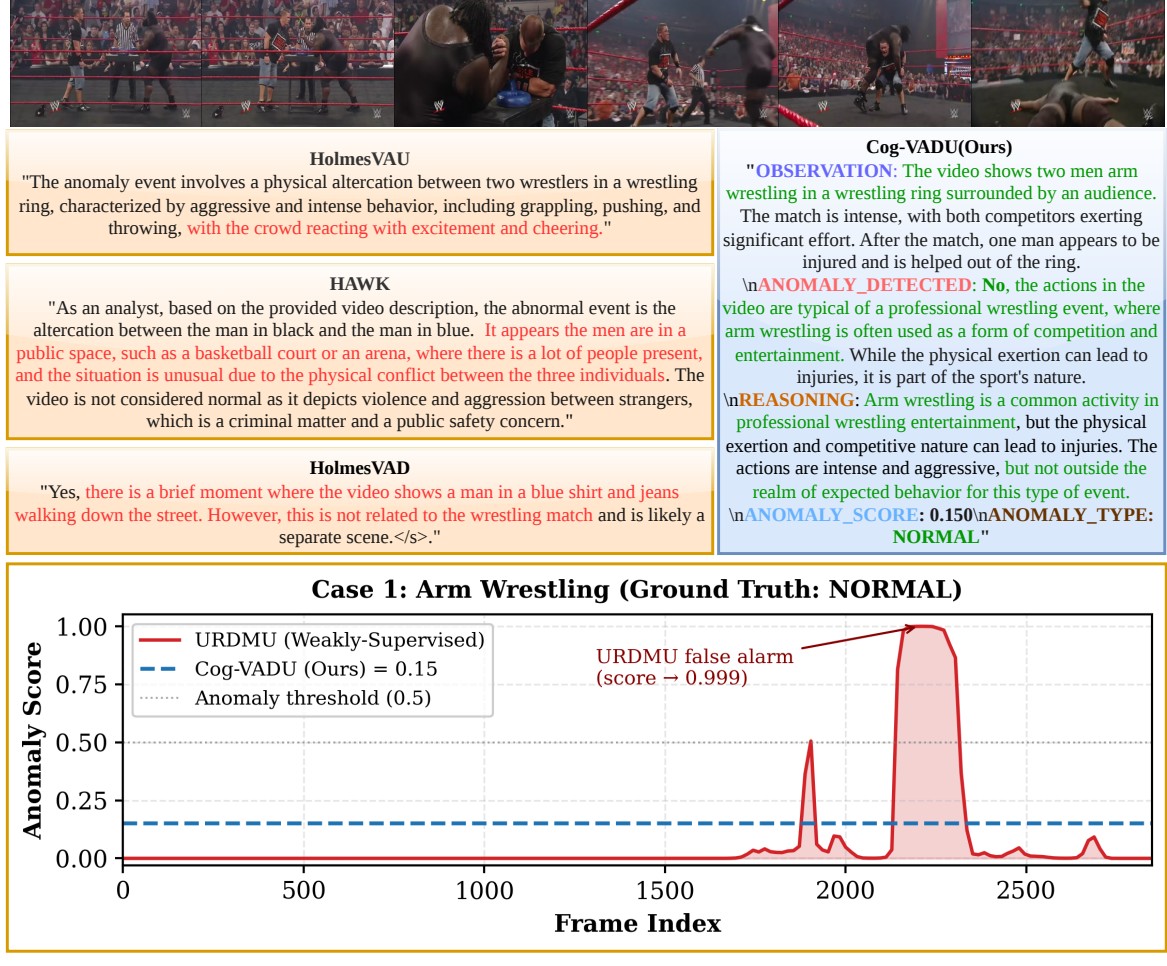

Figure 12: **Case 1 — Visually anomalous, semantically normal.** Comparison of four state-of-the-art VAD methods on an open-web arm wrestling video (ground truth: NORMAL). URDMU Zhou et al. (2023a) produces a strong false alarm (peak ≈ 0.999); HolmesVAD Zhang et al. (2024b) hallucinates an unrelated "man in a blue shirt" scene; HAWK Tang et al. (2024) fabricates a basketball court context; and HolmesVAU Zhang et al. (2025b) contradicts its own reasoning by labelling recognized wrestling activity as "anomaly events." In contrast, Cog-VADU correctly identifies the professional-wrestling context and outputs a calibrated score of 0.15 (NORMAL).

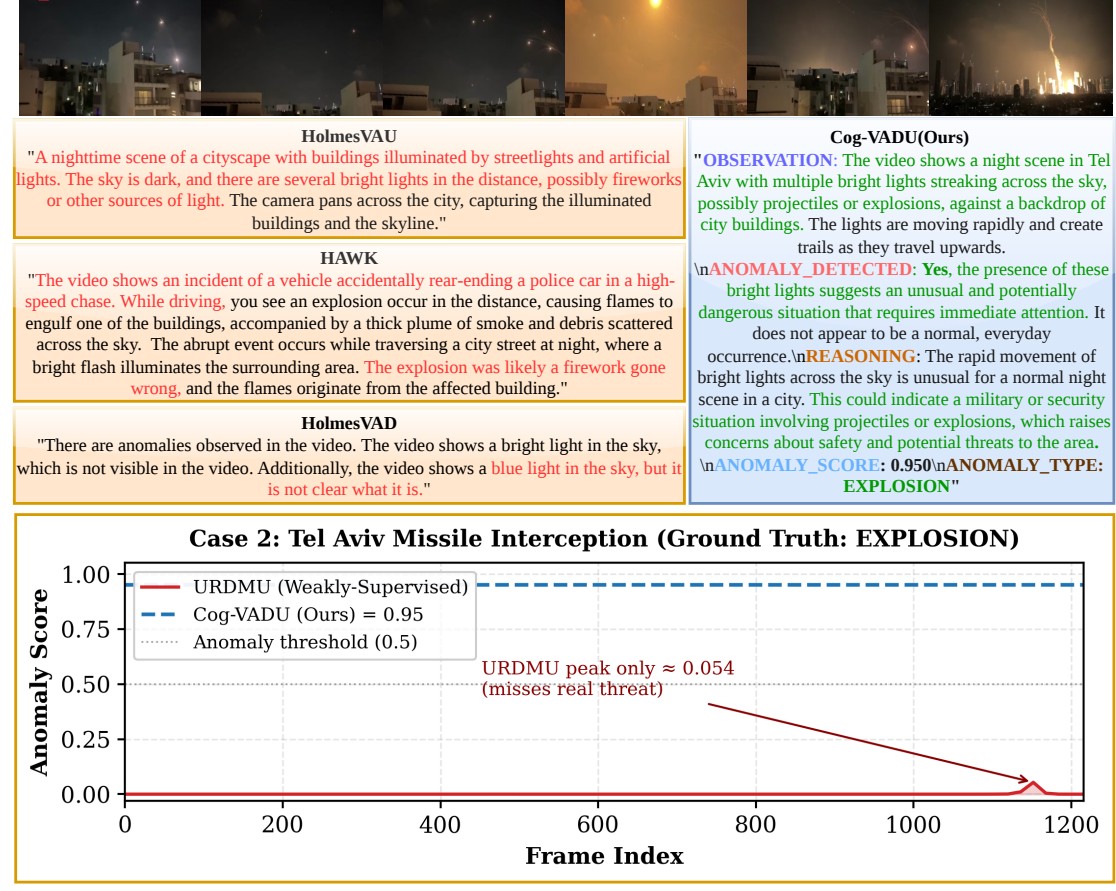

Figure 13: **Case 2 — Visually subtle, semantically anomalous.** Comparison on an open-web video of the Tel Aviv missile interception (ground truth: EXPLOSION). URDMU Zhou et al. (2023a) barely registers the event (peak ≈ 0.054); HolmesVAD Zhang et al. (2024b) hallucinates a "blue light in the sky"; HAWK Tang et al. (2024) confabulates a vehicle accident and "firework gone wrong"; and HolmesVAU Zhang et al. (2025b) misclassifies the projectiles as "fireworks or other sources of light."

### D.3 Failure Modes of Competing Methods

The two cases reveal three distinct failure modes in state-of-the-art baselines, demonstrating that current VAD methods fail in qualitatively different ways depending on their training paradigm:

1. **Pattern-matching failure** (URDMU Zhou et al. (2023a)): The weakly-supervised method scores the wrestling segment at $\approx 0.999$ purely from motion cues (false alarm), while completely missing the Tel Aviv event (peak $\approx 0.054$). This confirms that motion-statistical approaches are blind to semantic context.

2. **Rationale hallucination** (HolmesVAD Zhang et al. (2024b), HAWK Tang et al. (2024)): Holmes-VAD fabricates "a man in a blue shirt walking down the street" for the wrestling video, and HAWK confabulates "a vehicle rear-ending a police car in a high-speed chase" for the Tel Aviv footage — neither event exists. These are clear cases where training-based MLLMs produce confident but ungrounded explanations.

3. **Reasoning–decision misalignment** (HolmesVAU Zhang et al. (2025b)): For Case 1, HolmesVAU acknowledges the wrestling context yet still labels the events as "anomaly events," producing an internally contradictory output. For Case 2, it misclassifies the missile interception as "fireworks," demonstrating that training-induced priors override contextual evidence.

### D.4 A Documented Failure of Cog-VADU

While Cog-VADU's rationales are largely faithful to the visual evidence, we have identified a recurring *categorical labelling failure mode* that is informative for understanding the limits of the framework. Figure 14 illustrates this on an open-web video showing a sensory room being flooded with water.

In this example, all reasoning sub-steps are correct: the description accurately identifies the flooding event ("water starts flooding the room, causing objects to float and move around"), the reasoning properly recognises the safety concern ("not a normal activity and poses a safety concern"), and the anomaly score is calibrated high (**0.92**). However, the final anomaly *type* is misclassified as SUSPICIOUS instead of the more accurate FLOOD or ACCIDENT.

The root cause is structural rather than reasoning-related: our predefined anomaly database (Appendix C, Fig. 10) does not contain a FLOOD category, and the model gracefully falls back to the generic SUSPICIOUS class — precisely the behaviour encoded in our prompt for "unusual behaviour that raises concern but doesn't clearly fit other categories." This demonstrates two properties of Cog-VADU worth highlighting:

- **Graceful degradation:** The framework correctly detects, reasons about, and scores the anomaly even when its vocabulary is incomplete.

- **Diagnosable failures:** Unlike hallucinatory failures in baselines (Figures 12 and 13), this failure is interpretable and correctable simply by extending the anomaly database — without any retraining.

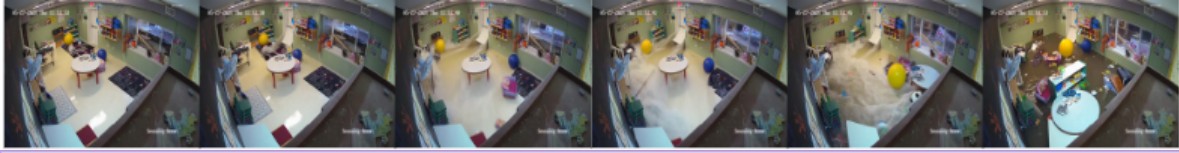

**Cog-VADU(Ours)**
"**OBSERVATION**: The video shows a sensory room that is initially empty and organized, with toys and furniture neatly arranged. Suddenly, water starts flooding the room, causing objects to float and move around, creating a chaotic scene. \n**ANOMALY_DETECTED: Yes**, The flooding of the sensory room is an anomaly that disrupts the normal use of the space and could potentially lead to accidents or injuries. \n**REASONING**: The video shows a sensory room that is initially empty and organized, with toys and furniture neatly arranged. Suddenly, water starts flooding the room, causing objects to float and move around, creating a chaotic scene \n**ANOMALY_SCORE: 0.920**\n**ANOMALY_TYPE: SUSPICIOUS**"

Figure 14: **Failure case — categorical labelling error under graceful degradation.** Cog-VADU is applied to an open-web video of a sensory room being flooded with water. The model produces a factually accurate *description* ("water starts flooding the room, causing objects to float and move around"), correctly identifies the *reasoning* ("not a normal activity and poses a safety concern"), and outputs a well-calibrated *anomaly score* of 0.92. However, the predicted *anomaly type* is SUSPICIOUS instead of the more precise FLOOD. The root cause is structural rather than reasoning-related: our predefined anomaly database does not contain a FLOOD category, and the model gracefully defaults to the generic SUSPICIOUS class. This failure is correctable in a training-free manner by simply extending the textual anomaly database, in contrast to opaque training-induced failures observed in baselines.

## D.5 Summary

Taken together, the three case studies provide direct evidence that: *(i)* Cog-VADU's rationales are grounded and consistent with predictions; *(ii)* the framework correctly disambiguates visually similar but semantically distinct events in both directions (suppressing false alarms on the wrestling case and recovering subtle anomalies on the Tel Aviv case); and *(iii)* competing training-based and pattern-based approaches fail in interpretable, diagnosable ways. The graceful-degradation behaviour observed in the flood case further indicates that Cog-VADU's failure modes are structural and addressable without retraining — a property not shared by competing supervised methods.

## E Error Propagation Analysis

A natural concern for any framework that propagates textual rationales across video segments is whether errors in early reasoning steps compound over time. This appendix presents a controlled rationale-corruption experiment that directly probes whether Cog-VADU exhibits error accumulation, along with an analysis of recovery dynamics and heavy-tailed cases.

## E.1 Experimental Setup

We sample 20 UCF-Crime videos uniformly at random from the test set. For each video, we execute two runs with *identical* visual inputs:

1. **Baseline run:** Standard Cog-VADU with no intervention.

2. **Corrupted run:** After segment 0 produces its own (correct) analysis, the temporal context $R_0$ passed to segment 1 is *replaced* with a deliberately incorrect rationale. From segment 1 onward, the model runs normally but with a contaminated reasoning chain.

To cover realistic failure modes observed in competing methods (Appendix D.3), we test three semantically distinct corruption types:

- **Wrong category:** A normal-scene context is replaced with a violent context (VIOLENCE, score 0.92).

- **Inverted:** An anomalous-scene context is replaced with a normal context (NORMAL, score 0.02).

- **Hallucinated entity:** A fabricated rationale describing a masked person with a weapon (ROBBERY, score 0.88).

For each segment $t$ we measure the absolute deviation

$$\Delta_t = |s_t^{\text{corrupt}} - s_t^{\text{baseline}}|, \tag{14}$$

and define *recovery* as the first segment $\tau$ where $\Delta_t$ falls below 0.10 and remains below that threshold for at least two consecutive segments. The choice of threshold 0.10 reflects the typical scoring jitter we observe between independent runs of identical configurations.

## E.2 Recovery Statistics and Propagation Curves

Table 12 summarises recovery statistics across the three corruption types, and Figure 15 plots the mean propagation curves.

Table 12: Recovery statistics under three rationale corruption types on 20 UCF-Crime videos. All 60/60 runs recover within the test horizon; the median recovery time is 1 segment for every corruption type.

| Corruption Type | Recovery Rate | Median $\tau$ | Mean Max $|\Delta s|$ | Mean Final $|\Delta s|$ |
|---|---|---|---|---|
| Wrong category (Normal $\rightarrow$ Violence) | 20/20 (100%) | 1 | 0.509 | 0.139 |
| Inverted (Anomaly $\rightarrow$ Normal) | 20/20 (100%) | 1 | 0.446 | 0.180 |
| Hallucinated entity (Masked + Weapon) | 20/20 (100%) | 1 | 0.512 | 0.142 |

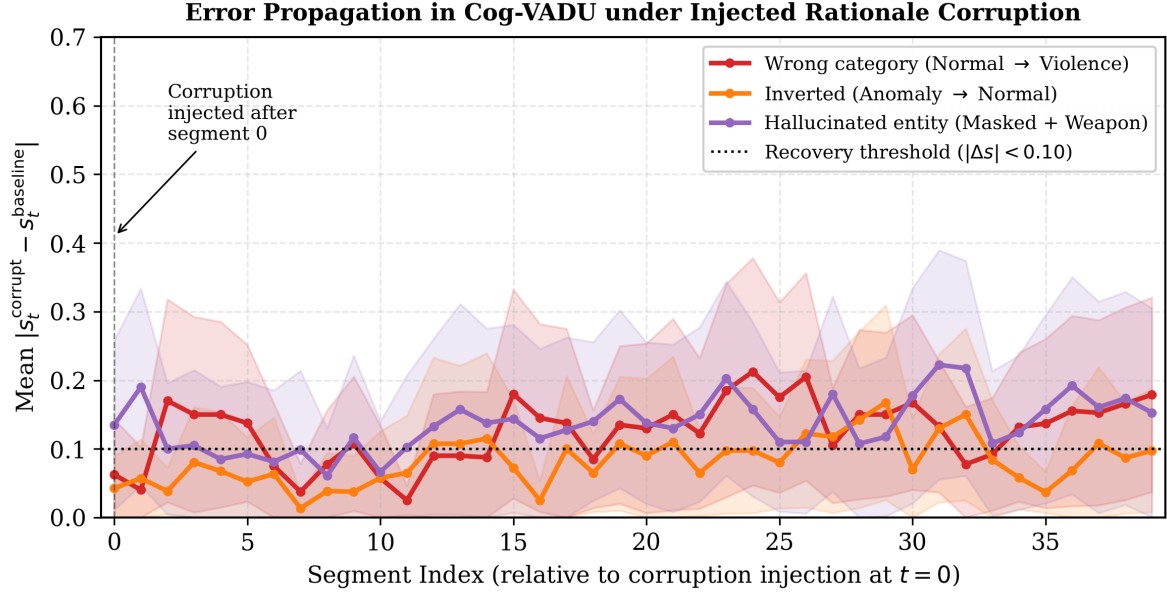

Figure 15: **Error propagation under injected rationale corruption.** For each of three corruption types, we plot the mean absolute deviation between corrupted-run and baseline-run anomaly scores at each segment index after the injection point ($t = 0$). Shaded bands denote $\pm 0.5\sigma$ across 20 UCF-Crime videos. Deviations remain bounded ($|\Delta s| < 0.10 - 0.20$) rather than accumulating upward, demonstrating that Cog-VADU's per-segment visual grounding prevents long-range error propagation.

## E.3 Key Findings

Three observations follow from the results above:

1. **Universal recovery.** Across all 60 video–corruption pairs (20 videos × 3 corruption types), Cog-VADU recovers in 100% of cases. The median recovery time is **1 segment** for every corruption type, indicating that the model self-corrects almost immediately after the injection point.

2. **No upward drift.** The mean deviation curves remain bounded throughout the entire 40-segment window (Fig. 15). Rather than compounding, deviations oscillate around the recovery threshold ($|\Delta s| 0.10 - 0.20$), comparable to the scoring jitter observed between independent runs of identical configurations.

3. **Visual re-anchoring.** The injected rationale is overridden in subsequent segments because each segment is grounded in fresh visual evidence and re-processed independently by the LVLM. The temporal context biases the reasoning but does not dictate it.

### E.4 Heavy-Tailed Cases

While the median recovery time is 1 segment across all corruption types, the HALLUCINATED ENTITY corruption exhibits a right-skewed distribution with mean recovery 5.4 segments and standard deviation 10.6. A small subset of videos therefore exhibits slower recovery than the median. On inspection, these cases correspond to videos in which the injected description ("masked person with a weapon") is *partially consistent* with the actual scene context — for example, low-light surveillance footage with occluded figures — making it harder for the model to contradict the injected context purely from visual evidence. This suggests that error recovery is fastest when the injected rationale sharply contradicts the visual content and slowest when ambiguous visual cues are compatible with the corrupted prior — a sensible and interpretable behaviour.

### E.5 Why Error Accumulation Is Bounded

We attribute the absence of error accumulation to two structural properties of Cog-VADU:

- **Per-segment visual grounding.** At every segment, the LVLM re-observes the raw frames and performs scene understanding from scratch (Sec. 3.4). The prior rationale enters the prompt as a textual hint, not as a constraint, so it can be contradicted by current observations.

- **Cross-modal re-ranking in the refinement stage.** Even when the textual rationale chain drifts briefly, the refinement stage (Sec. 3.5) re-anchors predictions to visual embeddings via FAISS-based retrieval, providing an independent correction signal that further suppresses any residual drift.

### E.6 Limitations

The analysis presented here demonstrates strong empirical robustness under a controlled 20-video pilot, but two practical limitations are worth noting. First, the experiment is conducted on UCF-Crime; very long-form surveillance footage ($> 10$ minutes) may exhibit dynamics that we have not stress-tested here. Second, recovery is fastest when visual evidence sharply contradicts the corrupted rationale, so ambiguous scenes may exhibit longer-lived drift, as illustrated by the heavy-tailed cases in Sec. E.4.

## F  Computational Cost on Long Videos

A natural concern is the computational cost of the multi-step reasoning framework, particularly on longer surveillance videos. This appendix reports inference cost for both modes in which Cog-VADU operates. All experiments were conducted on a single NVIDIA RTX 3090 GPU (24 GB VRAM, CUDA 12.6) using VideoLLaMA3-7B Zhang et al. (2025a) in `bfloat16` precision.

Cog-VADU supports two evaluation modes that have fundamentally different cost profiles:

### F.1 Video-Level Reasoning Mode

This mode is used for anomaly understanding metrics (BLEU, METEOR, ROUGE-L, BERTScore) on HIVAU-70k. The model uniformly samples a fixed number of frames $N_f = 16$ from the *entire* video — regardless of duration — and processes them in a single forward pass. Computational complexity is therefore approximately $\mathcal{O}(N_f)$ with $N_f$ fixed, making inference time approximately *constant* in video duration.

In practice, inference time remains nearly constant even for long videos. For representative HIVAU-70k samples ranging from $43.8\,\mathrm{s}$ to $119.1\,\mathrm{s}$, model inference required only 6–8 seconds, while total processing time including frame extraction remained below 15 seconds per video. This makes video-level reasoning practical for long-form surveillance footage: a 10-minute video and a 1-hour video incur essentially the same model-inference cost.

### F.2 Frame-Level Localisation Mode

This mode is used for temporal anomaly detection metrics (AUC, AP) on UCF-Crime and XD-Violence. For fine-grained localisation, every 16th frame is treated as a segment center, and a 10-frame window around it is analysed (Sec. 3.4). The complexity therefore scales linearly with video length, $\mathcal{O}(T)$, where $T$ denotes the number of frames in the video.

Inference is fully parallelizable across segments and across videos, so multi-GPU deployment scales linearly without coordination overhead.

### F.3 Discussion

The two modes reflect a practical trade-off between efficiency and temporal precision: *sparse global sampling* (16 frames per video) enables efficient reasoning over long videos at the cost of no temporal localisation, while *dense segment-based analysis* (every 16th frame as a center) provides accurate frame-level localisation at the cost of linear-in-duration runtime.

Two further properties bear on practical deployment. First, because Cog-VADU is *training-free* and uses a frozen LVLM, no amortised training cost is incurred — the numbers reported above are the *total* cost of deploying the system. Second, the segment stride $d$ in frame-level mode is a tunable parameter: deployments requiring near-real-time analysis can increase $d$ (sparser sampling) to trade temporal resolution for throughput. We leave a systematic study of stride-vs-accuracy trade-offs and backbone-size scaling (e.g., 7B vs. 2B variants) on long-form surveillance footage to future work.

## G Additional Experimental Results

We further evaluate our method on multiple widely used benchmarks, UCF Crime Sultani et al. (2019), XD Violence Wu et al. (2020), and VANE Bharadwaj et al. (2025).

### G.1 Quantitative Results: Average Precision

We additionally report Average Precision (AP), following the evaluation protocol used in prior work Wu et al. (2020). Table 13 compares Cog-VADU with state-of-the-art (SOTA) approaches on the UCF-Crime dataset Sultani et al. (2019). AP is particularly suitable for highly imbalanced datasets such as UCF-Crime, where anomalous events represent only a small fraction of the video timeline. As shown in the table, Cog-VADU achieves the highest AP among training-free approaches and surpasses several supervised baselines, demonstrating the effectiveness of reasoning-driven anomaly detection without task-specific training.

### G.2 Qualitative results: UCF and XD Violence

Figure 16 and 17 presents additional qualitative examples, with the top two rows taken from the UCF-Crime dataset and the two from the XD-Violence dataset Wu et al. (2020). As shown, whenever an abnormal event occurs, i.e., an activity that deviates from the normal environmental context, the corresponding anomaly

Table 13: Average Precision (AP %) on the UCF-Crime dataset Sultani et al. (2019). "Orig." denotes results evaluated on the original annotations, while "Ours-Ann." reflects performance on our re-annotated split. Colored sections distinguish Weakly Supervised Non-Explainable, Explainable, and Training-Free Explainable methods.

| Method | AP (Orig.) | AP (Ours-Ann.) |
|---|---|---|
| **Weakly Supervised Non-Explainable Methods** | | |
| UR-DMU Zhou et al. (2023a) [AAAI-23] | 35.59 | 50.84 |
| **Explainable VAD Methods** | | |
| VERA Ye et al. (2025) [CVPR-25] | 32.83 | 45.12 |
| Holmes-VAU Zhang et al. (2025b) [CVPR-25] | **36.24** | **56.49** |
| **Training-Free Explainable Methods** | | |
| LAVAD Zanella et al. (2024) [CVPR-24] | 27.27 | 38.38 |
| **Ours** | **38.08** | **58.03** |

score produced by our Cog-VADU framework increases significantly. In contrast, for normal regions or normal videos (highlighted in green on the right), the anomaly scores remain low. Moreover, the temporal summaries generated by our method align well with the relevant video frames, further demonstrating the effectiveness of Cog-VADU in capturing and localizing anomalous events.

### G.3 Qualitative Results on the VANE Benchmark

Figure 18 presents additional qualitative examples demonstrating the effectiveness of our method (Cog-VADU). In particular, we show results from the *SORA* subset of the proposed VANE benchmark, which contains AI-generated synthetic videos exhibiting subtle and unconventional anomalies (e.g., unusual creature appearances or unrealistic environmental events) that go beyond traditional surveillance scenarios. These videos are generated using SORA, a state-of-the-art text-to-video generation model, and often appear highly realistic, making the underlying anomalies difficult to detect even for humans. Automatically identifying such inconsistencies is therefore a challenging task and represents an important step toward detecting AI-generated or manipulated videos in real-world environments. Despite these challenges, our approach produces structured and grounded explanations that explicitly connect visual observations → cognitive anomaly reasoning → final decision concisely and coherently.

In contrast, HAWK Tang et al. (2024) and Holmes-VAU Zhang et al. (2025b) often fail to generate consistent reasoning chains, which can lead to incorrect predictions. VAD-R1 Huang et al. (2025), a reinforcement learning–based reasoning framework, produces the correct answer in this example but exhibits minor hallucinations in the reasoning steps.

Overall, these qualitative results highlight the advantage of Cog-VADU in producing accurate predictions together with concise and reliable reasoning.

## H Limitations and Future Directions

We introduce the task of *Video Anomaly Reasoning*, enabling Multimodal Large Language Models (MLLMs) to perform deeper semantic analysis of anomalous events in videos in a zero-shot setting. While Cog-VADU demonstrates promising generalization from anomaly detection to anomaly understanding and reasoning, several limitations remain. In particular, the multi-step reasoning process increases inference time and introduces additional computational overhead.

Future work will focus on improving efficiency through adaptive reasoning mechanisms and frame selection strategies that prioritize informative video segments. In addition, our findings highlight the importance of modeling temporal cause and effect relationships in anomaly detection. Motivated by this observation, we plan to develop a benchmark that explicitly captures such causal and temporal dependencies, providing a stronger evaluation platform for future research in video anomaly reasoning.

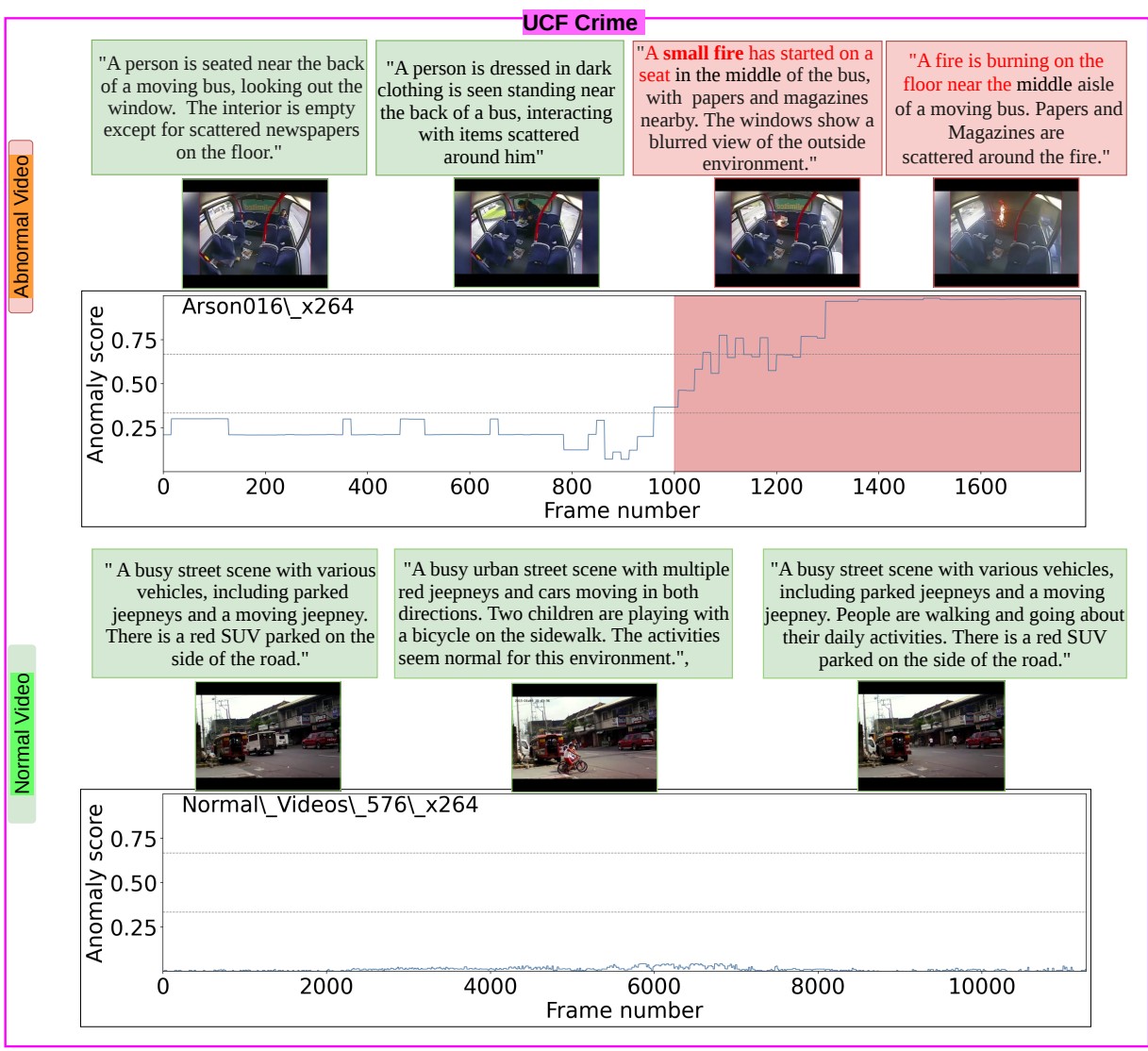

Figure 16: Qualitative results of our Cog-VADU method on two test videos, from UCF-Crime Sultani et al. (2019). For each video, plot the frame-wise anomaly scores and show keyframes with their most relevant temporal summaries **(green for normal, red for abnormal)**. These examples illustrate alignment between predicted anomaly scores, visual content, and descriptions. Ground-truth anomalies are highlighted in red.

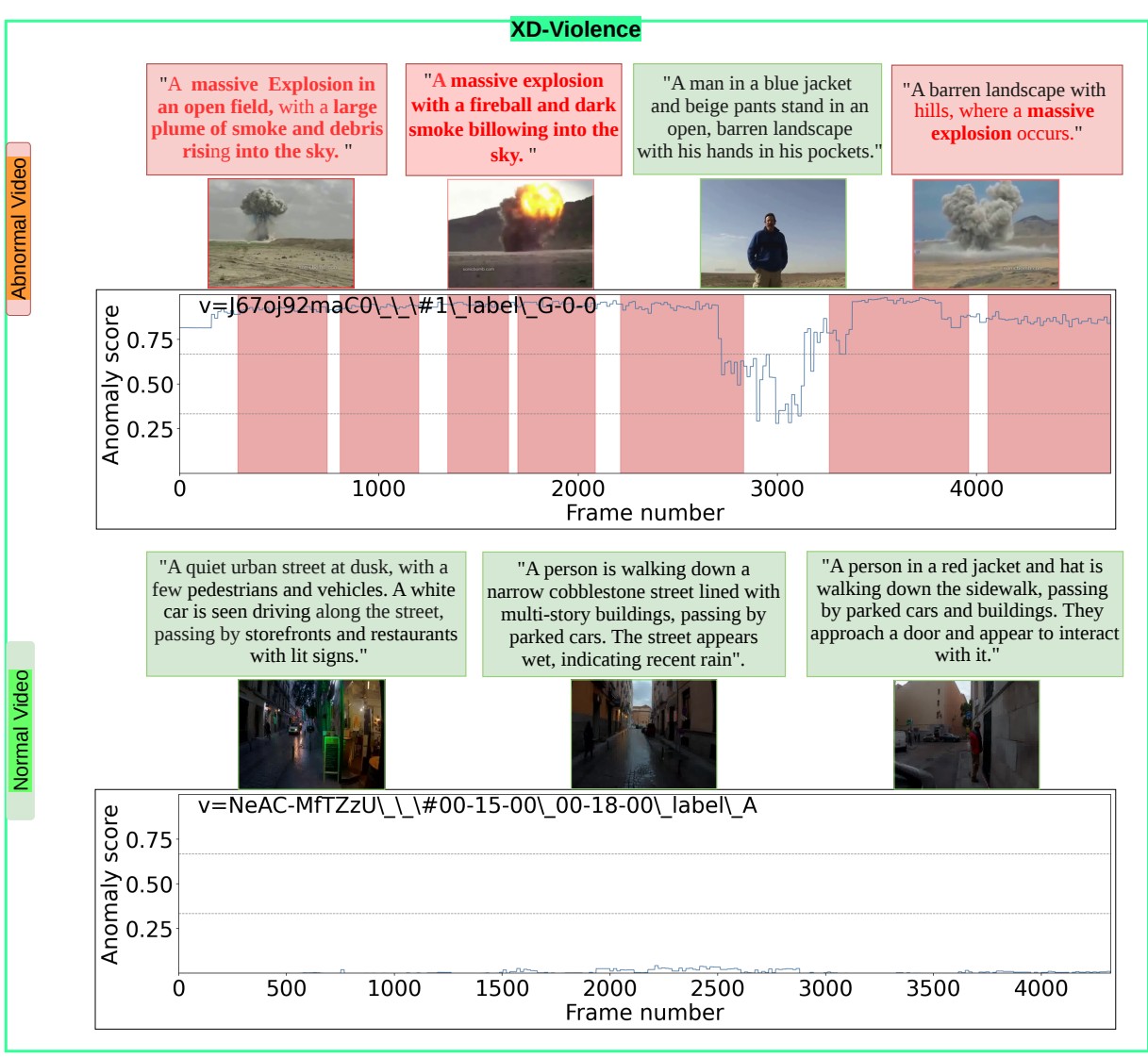

Figure 17: Qualitative results of our Cog-VADU method on two test videos, from XD-Violence Wu et al. (2020). For each video, plot the frame-wise anomaly scores and show keyframes with their most relevant temporal summaries **(green for normal, red for abnormal)**. These examples illustrate alignment between predicted anomaly scores, visual content, and descriptions. Ground-truth anomalies are highlighted in red.

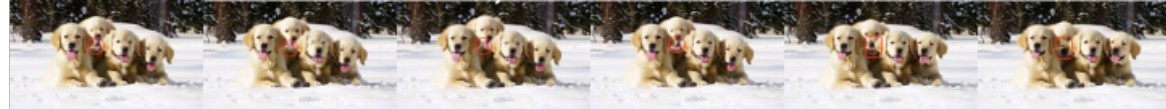

**"Question":** "What is the major inconsistency observed with the second puppy from the left in the video?

**Options:** A) The puppy's ears change color **B) The puppy's nose and tongue morph unnaturally**
C) The puppy's eyes change shape D) The puppy's fur changes texture",

**Cog-VADU(Ours)**
**OBSERVATION:** The video shows four golden retriever puppies lying in the snow. They are all facing the camera with their tongues out, appearing happy and relaxed. The puppy in the middle has a noticeable anomaly where its nose and tongue appear to morph unnaturally.\n**ANOMALY_DETECTED**: **Yes**\n**REASONING**: The second puppy from the left exhibits an unnatural distortion in its facial features, specifically the nose and tongue, which do not look like typical dog features. \nThis is the only anomaly detected in the video.\n**ANSWER: B**

**HolmesVAU**
The inconsistency exists, specifically between the second puppy from the left and the first puppy **D) The puppy's fur changes texture**

**HAWK**
A possible accurate description of the situation in the video might include: A group of dogs are shown in a snowy environment, including a young golden retriever wearing a yellow collar and a puppy wearing a pink collar. One of the puppies appears to be restless, as it is shown sitting and lying down on the snow. However, there is a notable inconsistency with one of the puppies that stands out from the others, as its ears change color, eyes change shape, and fur changes texture. This inconsistency could be a sign of a strange or abnormal behavior that might be the focus of attention in the video. **\"A) The puppy's ears change color.\\**

🟠Training based and reasoning Driven

🔵Training Free and reasoning Driven

**VAD-R1**
<think> The video shows a group of puppies in a snowy environment. The puppies are all facing forward, and their features, such as ears, eyes, and fur, appear consistent throughout the frames. There are no signs of unnatural changes in their appearance. The only anomaly observed is that the second puppy from the left has a slightly darker nose compared to the others, which could be considered a minor variation rather than a significant inconsistency.\n\n<think>Given the options, the most plausible inconsistency is related to the nose and tongue, as these features are often the most noticeable and can vary significantly. The other options, such as color change in ears or eyes, or changes in fur texture, do not seem to be present in the video </think>\n\n**n<answer>B</answer>**

Figure 18: Qualitative comparison on a video reasoning benchmark. Our training-free framework **Cog-VADU** (blue) correctly identifies the anomaly by reasoning over temporal visual inconsistencies, detecting that the second puppy's nose and tongue morph unnaturally across frames. In contrast, training-based VAD approaches (orange), including HAWK Tang et al. (2024), HolmesVAU Zhang et al. (2025b), and VAD-R1 Huang et al. (2025), either hallucinate unrelated changes (e.g., ear color or fur texture) or fail to localize the correct anomaly despite similar visual evidence. This example highlights the advantage of reasoning-driven anomaly detection in capturing subtle semantic inconsistencies that are not easily modeled by conventional training-based approaches.

