# OpenReview forum: "Cog-VADU: A Training-Free Cognitive Reasoning Framework for Video Anomaly Detection and Understanding"
_TMLR — Decision pending for TMLR_

### Review · Reviewer_dQie · 2026-05-12

**Summary Of Contributions:**

Summary

This paper proposes a novel training-free video anomaly detection method based on pretrained vision-language models.
The proposed method applies chain of thought prompt to extract anomaly score from video segments and then apply retrieval-based refinement to refine the initial score.
Further, the author re-annotates anomaly label to existing dataset since existing dataset sometimes lacks the label when anomaly happens.
The proposed method demonstrates better anomaly detection accuracy than existing supervised and non-supervised methods.


Strength

The author proposes a novel training-free video anomaly detection method by exploiting existing vision-language models.
To this end, the author proposes several

The author analyzes the bias in existing anomaly annotation and provides re-annotation that reduces the biases.

The proposed method demonstrates better anomaly detection performance on the re-annotated dataset.

The author provides several ablation study on the pretrained models and proposed modules.


Weakness

The proposed method demonstrates worse performance than existing training-based methods on original labels.
I want to see whether the performance of the proposed method increases when we modify the prompt or postprocess to mimic existing annotation bias (e.g. trigger-bias) discussed in Section 4.1.

It would be better to describe the annotation process in more detail. Is it manually annotated or by some VLM or so? Are there any mechanism to reduce bias or mislabel?

Further, I want to see the clarification whether existing training-based methods are trained with original label or re-annotated label. If the training annotations and test annotations are different, I think there are performance decrease and comparison become unfair.

The paper says the proposed zero-shot method is more generalizable to open-set scenario. I want to see the clarification whether the evaluation setting corresponds to this open-set scenario.

Resolutions of some figures are low.

**Audience:**

Yes

**Audience Explanation:**

The work would be interesting for those who work on video anomaly detection.

**Broader Impact Concerns:**

Already discussed after conclusion section.

**Claims And Evidence:**

Yes

**Claims Explanation:**

The proposed method demonstrates good performance on the re-annotated benchmark.

Further, the author conducts several ablation study to evaluate the proposed method.

**Requested Changes:**

Detailed explanation of the annotation process .

Clarification whether existing training-based methods are trained with original label or re-annotated label.
If the model is trained only with original label, I want to see the performance of the models trained with re-annotated label.

Clarification whether the evaluation setting corresponds to open-set scenario.

---

> ### Author Response · Authors · 2026-06-13
> **Author Response to Reviewer dQie**
>
> We thank Reviewer dQie for the positive assessment and the constructive suggestions. All corresponding changes are integrated into the revised manuscript (marked in blue). We address each point below.
>
> **1. Does mimicking trigger-bias improve performance on original labels?**
> Cog-VADU achieves stronger gains under the unified annotations than under the original labels, which directly supports our central observation: existing annotations are predominantly trigger-centric and often omit semantically meaningful effect-based or contextual anomaly states (Sec. 4.1). Reasoning-based models tend to detect such states, whereas the original annotations do not label them, so reasoning-driven methods are penalized under incomplete ground truth despite producing semantically consistent predictions. Rather than tuning Cog-VADU to reproduce this incompleteness, we strengthened the discussion in Sec. 4.1 and added qualitative examples where Cog-VADU detects temporally consistent anomaly continuations omitted in the original labels.
>
> **2. Annotation process manual or VLM? Bias-reduction mechanisms?**
> We expanded the annotation protocol (Appendix B). The unified re-annotations are fully manual, with no VLM or automatic relabeling at any stage a deliberate choice, since using a VLM to label data on which VLM-based methods are evaluated would risk circular evaluation. Each video was reviewed in multiple inspection passes. Bias is reduced by following a deterministic, rule-based protocol (label all frames depicting the cause, effect, escalation, or aftermath of an anomaly) rather than subjective per-frame judgement, which structurally reduces inter-annotator variance, and by keeping the protocol model-independent (no model predictions consulted).
>
> **3. Are training-based methods trained on original or re-annotated labels?**
> All training-based baselines use their original pretrained models (trained on original labels) and are evaluated on both the original and re-annotated test sets. We do not retrain any baseline on re-annotated labels, as retraining all baselines is computationally prohibitive, and we explicitly acknowledge that this places training-based methods at a disadvantage on the re-annotated labels. To ensure fairness, we report all results on both label sets (Tables 2 and 3), so each method can be assessed in its native setting and under the corrected annotations. This dual reporting is now stated explicitly in the revised paper.
>
> **4. Does the evaluation setting correspond to open-set?**
> Yes. Our open-set claim is supported by cross-dataset evaluation (Sec. 4.1–4.3), where Cog-VADU is evaluated on anomaly categories not seen during any training or prompt design. We additionally evaluate on anomaly types present in VANE-Bench and HIVAU-70K but absent from UCF-Crime and XD-Violence, demonstrating generalization to novel anomaly types without dataset-specific adaptation, and we show qualitative open-set cases in Appendix D. In contrast, the training-based baselines are tested on datasets sharing the same anomaly taxonomy as their training data  a closed-set setting.
>
> **5. Low figure resolution.**
> We have replaced the affected figures, like Fig. 2 and Fig. 3, with high-resolution versions in the revised submission.
>
> We thank the reviewer again and welcome any further discussion.

---

### Review · Reviewer_mpFC · 2026-05-26

**Summary Of Contributions:**

In this paper, the authors present Cog-VADU, a fully training-free framework for video anomaly detection and understanding based on large vision-language models (LVLMs). The key idea is to reformulate anomaly detection as a sequential reasoning problem rather than a pure visual recognition task. To achieve this, the authors introduce Chain-of-Anomaly Detection Thought Prompting (CoADTP), which guides an LVLM to perform structured reasoning over consecutive video segments through a chain of observations, descriptions, anomaly assessments, and severity scoring. The framework further incorporates a cross-modal re-ranking mechanism that aligns textual rationales with visual representations to improve temporal consistency and reduce hallucinations.

The paper evaluates Cog-VADU on multiple anomaly detection and anomaly understanding benchmarks, including UCF-Crime, XD-Violence, VANE-Bench, and HIVAU-70K. Experimental results suggest that the proposed framework achieves competitive zero-shot anomaly detection performance while generating interpretable reasoning outputs. The authors also demonstrate that CoADTP can improve the reasoning capabilities of multiple LVLM backbones without additional training.

**Additional Comments:**

**Questions:**

1. How sensitive is CoADTP to prompt wording and prompt structure? Have the authors evaluated prompt robustness?

2. Can the authors quantify the contribution of each component separately (reasoning chain, anomaly database, temporal feedback, cross-modal reranking)?

3. How does rationale propagation behave when an early reasoning step is incorrect? Is there evidence of error accumulation over time?

4. Have the authors evaluated reasoning faithfulness beyond lexical overlap metrics such as BLEU and ROUGE?

**Audience:**

Yes

**Audience Explanation:**

The topic is relevant.

**Claims And Evidence:**

Yes

**Claims Explanation:**

**Strengths:**

1. The paper reformulates video anomaly detection as a reasoning problem and leverages LVLMs for anomaly understanding rather than purely visual classification.

2. Cog-VADU does not require task-specific training or fine-tuning, making it computationally attractive and easy to deploy.

3. The proposed chain-of-thought prompting generates textual rationales that improve transparency compared to conventional anomaly detection methods.

4. Experiments on multiple anomaly detection and understanding benchmarks demonstrate competitive performance across several LVLM backbones.

**Weaknesses:**

1. The framework primarily combines existing components such as chain-of-thought prompting, temporal feedback, and retrieval-based refinement, making the conceptual novelty somewhat incremental.

2. The paper argues that reasoning improves anomaly detection but provides little evidence regarding the faithfulness, correctness, or robustness of the generated rationales.

3. The method relies heavily on carefully designed prompts and anomaly priors, yet robustness to prompt variations is not thoroughly studied.

4. The computational cost and effectiveness of the multi-step reasoning framework on long surveillance videos remain unclear.

5. More details on annotation quality and inter-annotator agreement would strengthen confidence in the newly introduced benchmarks.

**Requested Changes:**

See Weaknesses section.

---

> ### Author Response · Authors · 2026-06-13
> **Author Response to Reviewer mpFC**
>
> We thank Reviewer mpFC for the positive assessment and the thoughtful questions. The feedback prompted several new experiments and additions to the revised manuscript (all changes marked in blue). We address each question below, referencing the relevant sections of the revised paper.
>
> **Q1. Prompt sensitivity to wording and structure.**
> We added a prompt sensitivity analysis (Sec. 3.4, Table 1) comparing three prompts at different structural levels: a minimal caption-style prompt, a format-structured surveillance prompt (anomaly categories + output template but no reasoning), and our full CoADTP prompt. The gap between the minimal and format-structured prompts is small (+2.92% AUC, −0.31% AP), whereas CoADTP adds +7.77% AUC and +16.23% AP over the minimal prompt. This indicates the gains stem from *reasoning-level* structure rather than surface wording, and is reinforced by our cross-backbone results, where CoADTP improves six architecturally distinct LVLMs without prompt re-tuning.
>
> **Q2. Per-component contribution.**
> We substantially expanded the ablation (Table 9, Appendix C) to report each textual component *in isolation* above the base prompt, with score smoothing disabled. Each component independently improves over the base prompt, and they contribute along complementary axes: Reasoning is the strongest AUC driver (+4.37%), while Anomaly DB and Temporal Feedback are the strongest AP drivers (+12.42% and +11.53%). Cross-modal re-ranking adds a final corrective stage (+1.66% AUC), confirming the components are complementary rather than redundant.
>
> **Q3. Error propagation when early reasoning is incorrect.**
> We added a controlled rationale-corruption experiment (Appendix E). After segment 0, the temporal context is deliberately replaced with one of three corrupted rationales (wrong-category, inverted, hallucinated entity). Across all 60 video-corruption pairs (20 videos × 3 types), Cog-VADU recovers in 100% of cases with a median recovery of 1 segment, and deviations remain bounded rather than accumulating. We attribute this to per-segment visual grounding (each segment is re-observed from scratch) and the cross-modal refinement stage.
>
> **Q4. Faithfulness beyond BLEU/ROUGE.**
> We added a BERTScore evaluation (Sec. 4.2, Table 5) measuring semantic similarity via contextual embeddings. Cog-VADU attains an overall F1 of 0.316 without any task-specific training, outperforming the training-based explainable baseline Holmes-VAD (0.245). On the judgement task, which directly tests the anomaly decision, Cog-VADU achieves 0.545, second-highest among all methods and close to the in-domain-trained Holmes-VAU (0.652).
>
> *Qualitative faithfulness and failure analysis.* To complement these quantitative metrics, we added a qualitative study (Appendix D) on open-web videos outside the training distribution of all baselines. A symmetric pair of cases a visually-anomalous-but-normal arm wrestling match (Cog-VADU correctly scores 0.15, Normal) and a visually-subtle-but-anomalous missile interception (correctly 0.95, Explosion) demonstrates faithful, context-driven rationales. We also document a Cog-VADU failure (a flooded room mislabeled Suspicious due to a missing Flood category) and contrast it with three failure modes of competing methods (pattern-matching, rationale hallucination, and reasoning-decision misalignment).
>
> **On the listed weaknesses.**
>
> *Novelty (W1):* CoADTP's contribution is the integration of known ingredients into a recurrent reasoning chain for training-free VAD with cross-modal re-anchoring; the empirical insight that structured reasoning amplifies backbone capability holds across six backbones and is confirmed by the BERTScore analysis.
>
> *Rationale faithfulness/robustness (W2):* Addressed by the BERTScore evaluation (Q4), the qualitative faithfulness/failure study (Appendix D), and the error-propagation experiment (Q3).
>
> *Computational cost (W4):* We added an inference-cost analysis (Appendix F). Cog-VADU runs in two modes: video-level reasoning (16 frames, single pass, O(N_f)  constant in duration, 6–8s/video) and frame-level localization (per-segment, O(T) — linear in duration), making anomaly understanding directly applicable to long-form footage.
>
> *Annotation quality and inter-annotator agreement (W5):* We expanded the annotation protocol (Appendix B): fully manual (no VLM/auto-relabeling), multiple inspection passes, and bias-reduction safeguards. The re-annotation follows a deterministic rule-based protocol, labeling all frames depicting the cause, effect, escalation, or aftermath of an anomaly rather than subjective judgement, structurally reducing inter-annotator variance. All results are reported on both original and unified labels for transparency.
>
> We hope these additions address the reviewer's questions and welcome further discussion.

---

### Review · Reviewer_BhGB · 2026-05-30

**Summary Of Contributions:**

The actual contribution of Cog-VADU is that it reformulates VAD as a training-free sequential reasoning problem based on LVLMs, improves the interpretability and temporal stability of zero-shot anomaly detection through structured prompting and a simple cross-modal post-processing step, and highlights the cause-effect omission problem in existing annotations.

**Audience:**

No

**Audience Explanation:**

Although the paper addresses an interesting problem, I am not convinced that the findings would be sufficiently informative for the TMLR audience. The proposed framework mainly combines existing ingredients, including structured prompting, temporal feedback, and retrieval-based re-ranking, without introducing a technically distinctive contribution or a clearly reusable methodological insight. The practical recipe is also not described in a way that would make the method broadly adoptable, and the demonstrated utility does not appear strong enough to compensate for the limited novelty. Therefore, I believe the paper’s findings may be of limited interest beyond the specific experimental setting considered here.

**Broader Impact Concerns:**

Since this work targets video surveillance, it may raise concerns about privacy, false positives, over-surveillance, and potential bias. The paper should include a clearer broader impact discussion on responsible deployment and safeguards against misuse in real-world monitoring systems.

**Claims And Evidence:**

No

**Claims Explanation:**

The paper provides some evidence that Cog-VADU can improve zero-shot VAD performance and generate interpretable rationales. However, the evidence is not fully sufficient to support all of the paper’s stronger claims, especially regarding robust generalization, causal understanding, and the fairness of the unified re-annotation protocol. The proposed components appear to be a useful combination of structured prompting and cross-modal post-processing, but the experimental analysis does not fully disentangle their effects or establish that the method truly performs reliable causal reasoning.

**Requested Changes:**

The paper should provide a more detailed analysis of how the different components of the framework interact with each other, especially the connection between CoADTP, temporal feedback, cross-modal re-ranking, and the final score aggregation. This is critical for acceptance, because the current evidence does not sufficiently show whether the performance gain comes from structured reasoning, temporal feedback, retrieval-based smoothing, or their specific combination. More controlled ablations and diagnostic experiments would make the contribution clearer.

The paper should also strengthen the evidence for its explainability claims. This would strengthen the work, although I would not consider it by itself critical for acceptance. In particular, the authors should provide more diverse qualitative examples showing when the generated rationales are faithful, when they fail, and how they help distinguish visually similar but semantically different cases. Such examples would make the claimed interpretability more convincing and practically meaningful.

---

> ### Author Response · Authors · 2026-06-13
> **Author Response to Reviewer BhGB**
>
> We thank Reviewer BhGB for the detailed and constructive review. The comments on component disentanglement, explainability, and broader impact prompted substantial new experiments and additions to the revised manuscript (all changes marked in blue). We summarise our responses below, referencing the relevant sections of the revised paper.
>
> **1. Component interaction and controlled ablations.**
> We substantially expanded the ablation study (Table 9, Appendix C — Ablation on UCF-Crime) to disentangle each component. Beyond the cumulative ablation, we now report each textual component (Step-wise Reasoning, Anomaly Database, Temporal Feedback) *in isolation* above the base prompt, with score smoothing disabled:
> - Each component independently improves over the base prompt — none is redundant.
> - Contributions lie on complementary axes: Reasoning is the strongest AUC driver (+4.37%), while Anomaly DB and Temporal Feedback are the strongest AP drivers (+12.42% and +11.53%).
> - Cross-modal re-ranking provides a final corrective stage (+1.66% AUC).
>
> We additionally added a controlled error-propagation experiment (Appendix E) that probes how the components interact under perturbation: deliberately corrupting the temporal context yields 100% recovery within a median of 1 segment across 60 video-corruption pairs, showing the recurrent reasoning chain is self-correcting rather than error-accumulating.
>
> **2. Explainability: diverse qualitative examples.**
> We added a Faithfulness and Failure Analysis (Appendix D) with a symmetric pair of open-web cases: a visually-anomalous-but-normal arm wrestling match (Cog-VADU correctly scores 0.15, Normal) and a visually-subtle-but-anomalous missile interception (correctly 0.95, Explosion). These directly demonstrate faithful rationales and discrimination of visually similar but semantically different cases. We also document a Cog-VADU failure (a flooded room mislabeled Suspicious due to a missing Flood category) and contrast it with three distinct failure modes of competing methods (pattern-matching, rationale hallucination, and reasoning-decision misalignment).
>
> **3. Broader Impact.**
> We substantially expanded the Broader Impact statement to address all four concerns: privacy, false positives and over-surveillance, bias and fairness, and responsible deployment, with concrete recommendations (human-in-the-loop review, prohibition on predictive-policing use, and disclosure of documented failure modes).
>
> **On novelty**
> While CoADTP integrates known ingredients, its contribution is their integration into a *recurrent reasoning chain for training-free VAD* with cross-modal re-anchoring. The reusable empirical insight that structured reasoning amplifies backbone capability is demonstrated across six LVLM backbones and confirmed by a new BERTScore analysis (Sec. 4.2, Table 5), where stronger backbones cluster at 0.29–0.35 while weaker ones remain at 0.13–0.15.
>
> **On causal reasoning, generalization, and annotation fairness.**
> We clarify three points. First, on *causal reasoning*: we do not claim formal causal inference; Cog-VADU performs contextual and temporal reasoning over rationales. The error-propagation experiment (Appendix E) provides direct evidence that this reasoning is stable rather than fragile. Second, on *generalization*: our open-set claim is supported by cross-dataset evaluation (Sec. 4.1) on anomaly types absent from the training distribution, plus the qualitative open-web cases in Appendix D. Third, on *annotation fairness*: we expanded the annotation protocol (Appendix B) detailing the fully manual procedure, bias-reduction safeguards, and model-independence, and we report all results on *both* the original and unified labels so that no method is advantaged by the re-annotation.
>
> We hope these additions address the reviewer's concerns and welcome any further discussion.